# Simple physical mixing of zeolite prevents sulfur deactivation of vanadia catalysts for NO$_x$ removal

Inhak Song[1,5], Hwangho Lee[1,5], Se Won Jeon[1], Ismail A. M. Ibrahim [2,3], Joonwoo Kim[4], Youngchul Byun[4], Dong Jun Koh[4], Jeong Woo Han [2] & Do Heui Kim [1✉]

NO$_x$ abatement has been an indispensable part of environmental catalysis for decades. Selective catalytic reduction with ammonia using V$_2$O$_5$/TiO$_2$ is an important technology for removing NO$_x$ emitted from industrial facilities. However, it has been a huge challenge for the catalyst to operate at low temperatures, because ammonium bisulfate (ABS) forms and causes deactivation by blocking the pores of the catalyst. Here, we report that physically mixed H-Y zeolite effectively protects vanadium active sites by trapping ABS in micropores. The mixed catalysts operate stably at a low temperature of 220 °C, which is below the dew point of ABS. The sulfur resistance of this system is fully maintained during repeated aging/regeneration cycles because the trapped ABS easily decomposes at 350 °C. Further investigations reveal that the pore structure and the amount of framework Al determined the trapping ability of various zeolites.

[1] School of Chemical and Biological Engineering, Institute of Chemical Processes, Seoul National University, Seoul, Republic of Korea. [2] Department of Chemical Engineering, Pohang University of Science and Technology (POSTECH), Pohang, Gyeongbuk, Republic of Korea. [3] Department of Chemistry, Faculty of Science, Helwan University, Ain-Helwan, Cairo, Egypt. [4] Research Institute of Industrial Science and Technology (RIST), Gwangyang-si, Jeollanam-do, Republic of Korea. [5] These authors contributed equally: Inhak Song, Hwangho Lee. ✉email: dohkim@snu.ac.kr

Global energy consumption has increased over the past few decades, and catalytic processes are necessary to lower the levels of harmful pollutants emitted from the use of fossil fuels in industrial facilities and combustion engines. Nitrogen oxides ($NO_x$) are a pollutant present in combustion gases that have serious deleterious effects on the human body, and these species also participate in photochemical processes that result in smog and acid rain[1,2]. Nevertheless, $NO_x$ can be efficiently removed from exhaust gases by implementing selective catalytic reduction with $NH_3$ ($NH_3$-SCR)[3–5].

$TiO_2$-supported vanadia catalysts are the most common materials used in $NH_3$-SCR processes since they offer excellent denitrification ability and reasonable resistance to sulfur[6–10]. Commercially-available vanadia catalysts exhibit a relatively high general operating temperature, in the range from 300 to 400 °C. However, as environmental regulations become more stringent and applied to various fields in the future, the necessity of operating SCR at low-temperature below 250 °C becomes more important. For tail-end configurations in which the SCR reactor is placed downstream of precipitator or particulate control unit, the exhaust gas temperatures are usually below 200 °C, where the catalyst exhibits a much lower efficiency[11]. Thus, a reheating system is essential to achieve the optimum catalytic efficiency that meets stringent $NO_x$ regulations. However, a duct burner or an electric heater used to intentionally raise the off-gas temperatures consumes additional fuel, and produces $NO_x$ and additional carbon dioxide. For example, about 0.5–1.5% of the total power generation in power plants is used to raise the temperature of the exhaust gas and operate the SCR system[12]. Hence, developing a low-temperature $NH_3$-SCR technology is a key technical, economic, and environmental challenge that must be overcome.

Many catalysts with superior low-temperature $NH_3$-SCR activity have been developed, such as Cu-zeolites or Mn oxides, but they are all not usable in most off-gas conditions because the active sites are severely deactivated by chemical poisoning with sulfur dioxide[13–16]. Recently, much effort has been made to solve the problem of sulfur deactivation in the SCR catalysts, for example, R.Yu et al. developed Cu-SSZ-13 zeolite-metal oxide hybrid catalyst that shows enhanced $SO_2$ tolerance by preferentially forming Zn sulfate over Cu sulfate[17], and L. Han et al. discovered that a mesoporous $TiO_2$ shell can improve the $SO_2$ resistance of $Fe_2O_3$ catalyst[18]. Unfortunately, however, it is difficult to commercialize in the field because the method of preparing catalyst is complicated and a very high temperature (650 °C) is required to regenerate the deactivated catalysts. Thus, to date, the only feasible option for efficient, low-temperature $NO_x$ removal is to increase the number of active V sites on the V-based catalysts that are not chemically deactivated with sulfur dioxide. However, even with increasing vanadia loading, V-based catalysts are not free from sulfur deactivation due to the formation of ammonium bisulfate (ABS) (Fig. 1a). ABS has a dew point typically between 280 and 320 °C and a melting point around 150 °C, and this condensed liquid ABS can physically block the pores in the catalyst, degrading the catalytic performance[19–21].

$$NH_3(g) + H_2O(g) + SO_3(g) \rightarrow NH_4HSO_4 \quad (1)$$

Equation (1) shows the formation of ABS. It originates from sulfur trioxide, which is formed by the oxidation of $SO_2$ catalyzed on the high loading vanadia material[22]. Thus, the current technology has a huge dilemma in that increasing V sites can improve the low-temperature activity[9,23] but also promote the formation of ABS, resulting in more rapid deactivation of the catalyst. Sulfur-trapping materials, such as $CeO_2$, were proposed to capture the sulfur as a metal sulfate but their protection abilities were insufficient to be implemented in the real-world off-gas conditions. In addition, they are non-renewable systems because the formed metal sulfates usually decompose above 700 °C, at which temperature vanadium oxides are severely sintered[24].

Here, we report that H–Y zeolite can be physically mixed with the vanadia catalyst to effectively trap liquid ABS, protecting most V sites from deactivation by ABS and demonstrating stable $NH_3$-SCR performance at 220 °C. The advantage of this system is that, unlike with stable metal sulfates, the unstable ammonium sulfate (ABS) salts can be decomposed at low temperatures, making the system reusable upon regeneration at temperatures as low as 350 °C. In addition, simple physical mixing approach is a cost-effective and practical solution that can be easily applied to the industry[25]. This catalyst-protection system enables low-temperature operation of $NH_3$-SCR in industrial plants, which has been impossible due to the above-described stability issues. It reduces additional fuel costs and carbon dioxide emissions by lowering the operating temperature and prolonging the regeneration cycle, consequently, contributing both significant economic and environmental benefits.

## Results and discussion

**Superior sulfur resistance of vanadia-zeolite mixed catalyst**. A supported vanadia on tungsta-titania catalyst (VWTi) with 5 wt.% of $V_2O_5$ was used in this work (see Supplementary Fig. 1 for the characterization of the catalyst). A physical mixture of VWTi and H–Y zeolite with a $Si/Al_2$ ratio of 12 (VWTi + Z) was prepared by mechanical mixing in a mortar, and the mass ratio of VWTi and zeolite in the mixture was 2:1. The laboratory reaction system simulated the off-gas emitted from a sintering plant by containing 500 ppm NO, 5% $CO_2$, 10% $H_2O$, 30 ppm $SO_2$ and 10% $O_2$ balanced with $N_2$, while 600 ppm $NH_3$ was introduced as reductant. The operating temperature (220 °C) was set below the dew point of ABS, and the space velocity was 150,000 mL/h·g catalyst to simulate ABS deactivation. Under these conditions, lab-made Cu-SSZ-13 and $Mn/TiO_2$ catalysts showed ~90% $NO_x$ conversion that completely deactivated in 2 h due to the formation of copper and manganese sulfates (Fig. 1b). The VWTi catalyst with high V loading (containing 5 wt.% $V_2O_5$) showed a moderate activity of ~65% conversion, and this is a fairly good low temperature performance compared to conventional catalyst with 3 wt.% $V_2O_5$ loading, which only exhibits ~30% conversion (Supplementary Fig. 7). However, the activity of VWTi catalyst gradually declined to below 50% after 22 h, reflecting the gradual deactivation by ABS explained above. Surprisingly, the VWTi + Z retained its activity above ~65% conversion after $SO_2$ aging for 22 h. We confirmed that physically mixed H–Y zeolite itself did not participate in $NH_3$-SCR reaction, did not capture $SO_2$, and there was hardly a change in the SCR reactivity of the VWTi catalyst (Supplementary Figs. 2 and 3a), but it only prevented ABS deactivation. It was also confirmed that H–Y zeolite can prevent ABS deactivation even with much higher $SO_2$ concentration of 100 ppm (Supplementary Fig. 4). The activity of VWTi + Z recovered almost completely after regeneration at 350 °C where ABS decomposes, as in the case of VWTi (Fig. 1c and Supplementary Fig. 5). Regeneration gas contains 10% $O_2$, 5% $CO_2$, and 10% $H_2O$ balanced with $N_2$. We directly observed desorption of sulfate species released from $SO_2$-aged VWTi catalyst during the regeneration step (Supplementary Fig. 6). Furthermore, VWTi catalyst with a much lower amount of V (3 wt.% $V_2O_5$) was also compared with its mixture with zeolite under $SO_2$ aging condition (Supplementary Fig. 7), verifying that the mixed zeolite can prevent ABS deactivation regardless of V loading.

A series of $SO_2$ aging for 22 h and subsequent regeneration was repeated 3 times over VWTi + Z, which verified the reusability of this system (Fig. 1c). The deactivation rate of VWTi + Z was maintained at about 1/3 of VWTi alone, even in multiple reaction

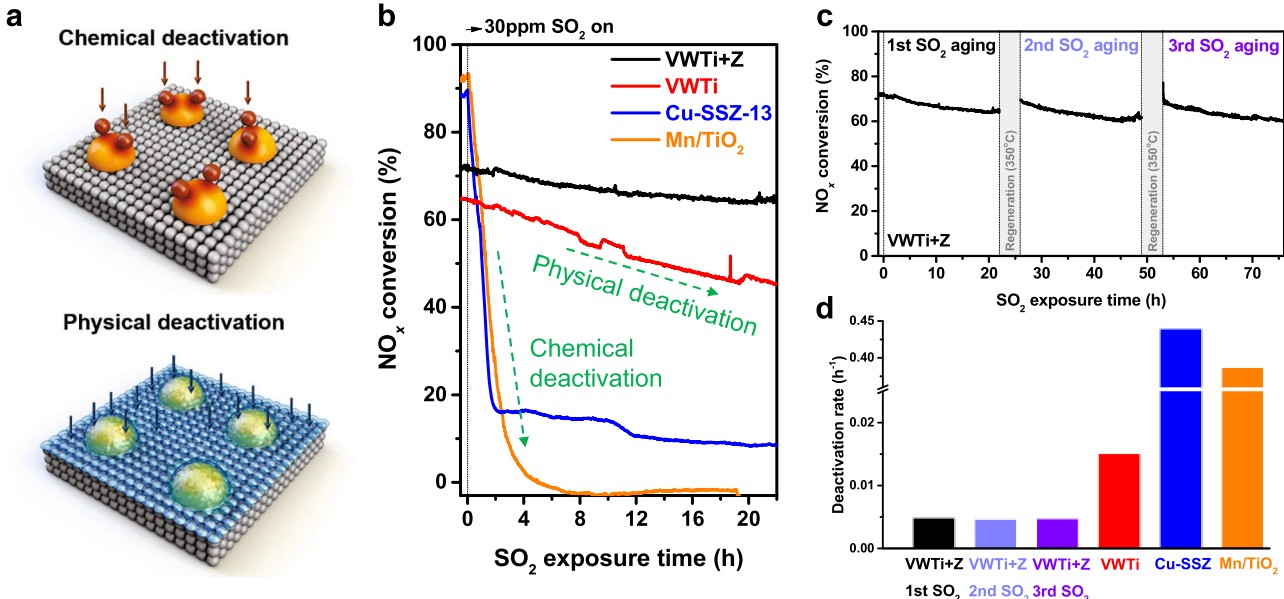

**Fig. 1 Stable and reusable NO_x removal system during NH₃-SCR reaction with SO₂. a** Schematic of two types of deactivation by sulfur species.
**b** Comparison of SO₂ resistance of VWTi + Z catalyst to VWTi and common low temperature catalysts (Cu-SSZ-13 and Mn/TiO₂). The catalysts
were aged under 500 ppm NO, 600 ppm NH₃, 10% O₂, 5% CO₂, 10% H₂O, and 30 ppm SO₂ balanced with N₂. **c** Regeneration and reusability test of the
VWTi+Z system until the third operation. Regeneration gas contains 10% O₂, 5% CO₂, and 10% H₂O balanced with N₂. **d** Deactivation rate of the catalysts
during SO₂ aging. Deactivation rates of the VWTi and VWTi+Z were obtained by linear fits of 22 h aging profiles and those of the Cu-SSZ-13 and Mn/TiO₂
were calculated from initial 2 h aging data.

tests, which was superior to the common low temperature catalysts by two orders of magnitude (Fig. 1d). The amount of deposited sulfur species over the catalysts was analyzed after ABS deactivation based on the mass of the VWTi to understand the role of H–Y zeolite (Supplementary Table 1). Although the activity of the VWTi + Z did not decrease considerably, unlike VWTi, there was no decrease in the deposited amount of sulfur on VWTi + Z immediately after reaction, which means that the physically mixed zeolite neither deterred the formation of ABS nor decomposed it.

**Post mortem examination of catalysts after sulfur aging**. Information on the local distribution of sulfur over post-reaction catalysts was obtained with a TEM-EDS analysis. The catalysts were dispersed on a TEM grid using acetone so that the solvent did not dissolve ABS. For aged VWTi catalysts, the distribution of sulfur appeared to resemble that of V and Ti (Fig. 2a, b), indicating that formed ABS uniformly distributed across the catalyst surface. For the case of aged VWTi + Z, however, most of the sulfur is located on the zeolite domains and not on VWTi (Fig. 2c, d). These results clearly demonstrate that the locations where sulfur species initially formed were different from the regions where they were deposited in the VWTi + Z. These phenomena do not result from a simple diffusion process of the sulfur species because (i) nearly no sulfur remained on VWTi compared to zeolite regions in the VWTi + Z mixed catalyst, and (ii) almost no change in ABS location was observed for aged VWTi+Silica mixtures, which showed no improvement in SO₂ resistance (Supplementary Fig. 8). Such observations allow us to propose that physically mixed zeolites absorb liquid ABS initially formed by and then initially covering VWTi during SO₂ aging, resulting in the protection of the active vanadia sites from poisoning.

Ar adsorption-desorption was also used to monitor changes in the micropore volume of the zeolite after ABS deactivation (Fig. 2e). After SO₂ aging for 22 h, the maximum value of the pore size distribution curve near 0.8 nm shrinks slightly due to pore

filling with ABS, as evidenced by the decrease in dV/dW from 0.40 to 0.32. The regeneration step fully restored the curve to the initial state, and subsequent 2nd and 3rd aging tests resulted in exactly the same behavior as the 1st aging. The complete regeneration of the micropore volume in the zeolite enabled essentially the same ability of the catalyst for ABS absorption during multiple operation, demonstrating the reusability of the catalyst system.

To directly observe the decomposition of adsorbed sulfur species into SO₂, multiple-aged VWTi + Z catalysts were heated to 900 °C in N₂ (Fig. 2f). It is interesting to note that, for the case of the 1st aged VWTi+Z, most of the SO₂ was desorbed at ~700 °C which originates from the decomposition of aluminum sulfate formed on the H–Y zeolite (Supplementary Fig. 3b). Such aluminum sulfate species might form during the decomposition process of ABS on the H–Y zeolite. As aging was repeated, the peaks at 700 °C became saturated, and the peaks at 440 °C originating from the decomposition of ABS on the zeolite were found to increase. (This temperature is slightly higher than the ABS decomposition temperature on VWTi (415 °C) and may be due to a different pore geometry. Also, ammonium bisulfate species decompose much more easily under regeneration conditions (H₂O, O₂, CO₂/N₂) than under N₂ conditions, so the catalyst can be regenerated at 350 °C (Supplementary Fig. 6). Thus, multiple operations of the system increased the amount of residual sulfur bound to the Al sites (Supplementary Table 1). In spite of the presence of residual sulfur, the ability of the zeolite to deter deactivation of the catalyst was confirmed not to deteriorate upon on repeated operation (Fig. 1c). To verify the negligible effects of residual sulfur on the ABS trapping ability of the zeolite, sulfur-saturated Y zeolite (S:Al = 1.4) was prepared and tested (Supplementary Fig. 9). Sulfur-saturated zeolite was confirmed to have similar ABS trapping ability with fresh zeolite, demonstrating the overall reusability of the VWTi + Z system regardless of the residual sulfur. In addition, the aging experiment was repeated by increasing the SO₂ concentration from 30 to

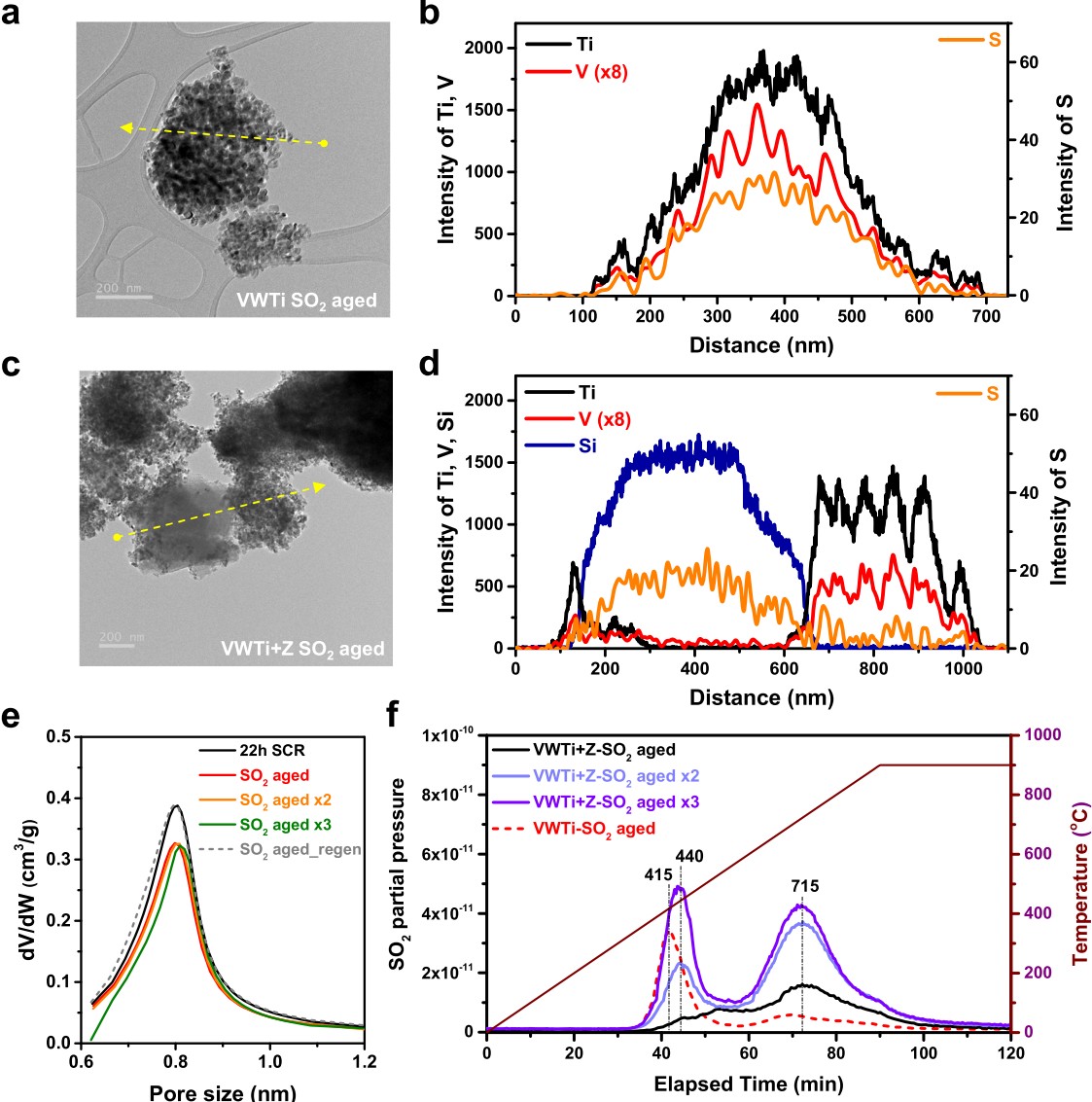

**Fig. 2 Location and species of sulfur after SO$_2$ aging. a, b** TEM image and line-EDS of the SO$_2$ aged VWTi. **c, d** TEM image and line-EDS of the SO$_2$ aged VWTi + Z. In the segregated region of the VWTi (Ti, black) and Y zeolite (Si, navy), a distribution of sulfur (orange) could be investigated. **e** Micropore distribution of the VWTi+Z fresh, SO$_2$-aged and regenerated catalysts from the Ar adsorption results. To eliminate the effect of the structure degradation under SCR conditions at 220 °C, VWTi + Z treatment under a NH$_3$-SCR reaction for 22 h (black) was suggested as a standard. **f** Thermal decomposition profiles of sulfur species on the SO$_2$-aged VWTi + Z and VWTi catalysts. Decomposed sulfur species were measured by using mass spectrometry, and any other sulfur species was not detected except SO$_2$. A low temperature peak at ~400 °C is assigned to SO$_2$ from the decomposition of ABS, and high temperature peak at ~700 °C is SO$_2$ from a decomposition of metal sulfate.

100 ppm for VWTi + sulfur-saturated zeolite to confirm whether the regeneration is still possible (Supplementary Fig. 10). Initial catalytic activity and lowered deactivation rate were also maintained after regenerating catalyst at 350 °C, indicating that ABS sorption function of zeolite still works even after saturation with sulfur.

**Identification of absorptive protection mechanism.** A migration model is suggested to explain the disparity between the location of ABS formation and deposition for the mixed catalysts. To observe the migration of ABS from the VWTi to the H–Y zeolite, ABS was pre-impregnated onto VWTi (ABS/VWTi) and then physically mixed with the H–Y zeolite. As a comparison, a sample was prepared without close physical contacts between the ABS/ VWTi and H–Y zeolite particles (ABS/VWTi + Z PM L). For the

case of ABS/VWTi + Z PM L, ABS on the VWTi could not migrate to the H–Y zeolite because of the separation between the VWTi and the zeolite domain[26]. Arrhenius plots of NO$_x$ removal rates at temperatures between from 135 and 215 °C were obtained in order to better understand the deactivation arising from a phase transformation of impregnated ABS from solid to liquid (Fig. 3a and Supplementary Fig. 11a). The slope in the Arrhenius plot for the ABS/VWTi+Z PM L material started to lower above 160 °C (Fig. 3a red curve), indicating that physical deactivation presumably occurs above that temperature due to phase transformation of ABS into liquid. However, ABS/VWTi + Z showed a linear Arrhenius plot without any deactivation, indicating that the VWTi was not deactivated by ABS in this temperature range (Fig. 3a black) even though it was pre-impregnated by ABS (2 wt.%). Above the melting point of ABS (160 °C), VWTi + Z and VWTi + Z PM L show same slopes of Arrhenius plots while

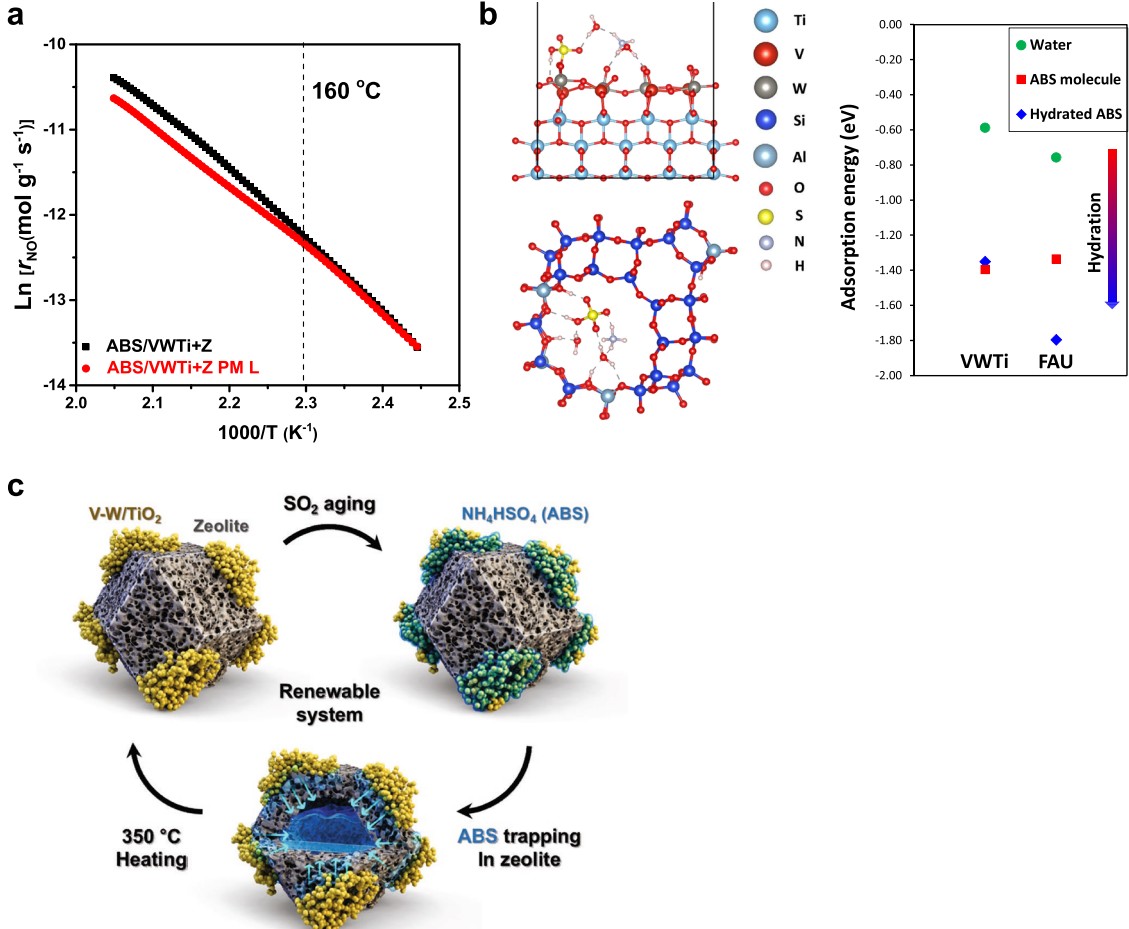

**Fig. 3 ABS trapping model by liquid phase migration. a** Arrhenius plots of ABS/VWTi + Z (black dot) and ABS/VWTi+Z PM L (red dot) during transient NH$_3$-SCR reaction (temperature from 100 °C to 220 °C with ramp rate 1 °C/min, Reaction gas contains 500 ppm NO, 500 ppm NH$_3$, 10% O$_2$, 5% CO$_2$, and 10% H$_2$O balanced with N$_2$; NO$_x$ conversion profiles in Supplementary Fig. 11a). **b** DFT calculation results regarding the stability of hydrated ABS by two H$_2$O molecules on the VWTi and H–Y zeolite. **c** Schematic illustration of an operating principle of the ABS trapping in the VWTi + Z system.

only pre-exponential factor decreased in the PM L sample, which clearly illustrates that the ABS gives rise to physical deactivation in the PM L catalyst (Supplementary Table 2). It can be inferred that pre-impregnated ABS did not remain on VWTi, but instead migrated to the H–Y zeolite through close physical contact between the VWTi and H–Y zeolite. Note that we conducted same experiments with much higher amount of pre-impregnated ABS (10 wt.%) on VWTi (Supplementary Fig. 12), and the difference between the two samples was remarkably observed. Since this migration is not a simple diffusion process as mentioned above, it is attributed to the ABS trapping ability of the H–Y zeolite as a result of its affinity to ABS, which protects VWTi from sulfur poisoning. Also, SO$_2$ deactivation behavior of physically mixed VWTi + Z catalyst was compared with that of loose-contacted VWTi + Z PM L sample (Supplementary Fig. 11b). It can be seen that VWTi + Z PM L sample shows a similar deactivation rate to that of pure VWTi, demonstrating that gaseous SO$_3$ cannot be captured directly by zeolite. This result suggests that mixed H–Y zeolite could not work to alleviate sulfur deactivation without close physical contact to VWTi, likely because the H–Y zeolite traps sulfur via liquid phase ABS, not by gas phase SO$_2$ or SO$_3$.

A computational comparison of the affinity of the two materials for ABS was carried out using a simple adsorption model of hydrated ABS on the VWTi surface and H-Y zeolite (Fig. 3b). Although the adsorption energies of ABS on the VWTi

and H-Y zeolite showed little difference, the adsorption energy strength for hydrated ABS by two H$_2$O molecules on H–Y zeolite ($E_{ads} = -1.79$ eV) was much higher than that on the VWTi surface ($E_{ads} = -1.35$ eV). Water molecules in the faujasite structure of H–Y zeolite enhanced the stability of ABS through the interaction with Brønsted acid sites generated by the framework Al, which rationalizes the migration of ABS between particles. Notably, a higher stability of ABS on the H–Y zeolite was also maintained in the adsorption of ABS hydrated by one and three H$_2$O molecules (Supplementary Fig. 13). Moreover, in presence of ammonia, the stability of ABS on H–Y zeolite was also much higher than that on the VWTi surface (Supplementary Fig. 14). To sum up the process (Fig. 3c), ABS is formed on the surface of VWTi via SO$_2$ oxidation under NH$_3$-SCR conditions with SO$_2$. However, this formed ABS cannot deactivate the vanadia catalyst because the physically mixed H–Y zeolite absorbs the ABS from the VWTi due to its excellent ABS trapping ability. Such absorptive protection behavior of the catalyst originates from the different stability of hydrated ABS on the VWTi and H–Y zeolite, which can be confirmed by the adsorption energy of hydrated ABS obtained from the DFT calculations. One question is how sulfur-saturated zeolite can trap ABS, as observed above, where the framework Al has a strong interaction with sulfate groups. We observed that the amount of Lewis acid sites decreases slightly after sulfation, but little change in the amount of Brønsted acid sites (Supplementary Fig. 15). This result clearly

demonstrates that the amount of Brønsted acid sites that play an important role in ABS migration is not changed by interactions with sulfate groups. It might be because the sulfated sites also act as the Brønsted acid sites, a well-known chemistry in sulfated metal oxides[27]. Our concept in this study is quite different from the existing sulfur trap systems where the sulfur has been trapped as a metal sulfate that is very difficult to decompose[28,29]. By instead trapping the sulfur species in the ammonium form, the mixed catalyst system can be regenerated at 350 °C, a significantly milder condition than the regeneration temperatures required for metal sulfates. After thermal treatments at 350 °C, the blocked pores of the zeolite are fully restored, so the system shows complete reusability during repeated operations (Figs. 1c and 2e). Our results imply that physical mixing can be an effective and facile solution to prevent the degradation of catalysts by promptly trapping deactivation species formed as byproducts during reaction.

**Factors determining the ABS trapping ability of zeolites.** To identify the factors that determine the ABS trapping ability, $SO_2$ stability tests were performed with various zeolites that have different $Si/Al_2$ ratios and structures (Supplementary Fig. 16). Deactivation rates decreased with a decrease in the $Si/Al_2$ ratio, and they decreased in the order of CHA, MFI and FAU with similar $Si/Al_2$ ratios but having small, medium and large pores, respectively (Supplementary Fig. 17). These results suggest that the ABS trapping ability of the zeolite is strongly dependent on the amount of framework Al and the zeolite pore structure— especially the pore size. With these two factors in mind, we hypothesize that the rate of ABS migration is a first-order function of the amount of framework Al ($C_{Al0}$), and we propose a mechanism of the migration by introducing several assumptions (the details of each of the assumptions are in the "Methods" section). According to the proposed model, the logarithm of the deactivation rate should be proportional to $C_{Al0}$, and a linear correlation could be confirmed from the first-order regression of the logarithm of deactivation rates as a function of the amount of the framework Al (Fig. 4).

The rate of the ABS migration is dependent on the zeolite structures, as evidenced by the different slopes of the linear fits ($k_m$: rate constant of ABS migration) for the various types of zeolite (CHA, MFI, and FAU). A large pore zeolite (FAU) has a larger $k_m$ value (0.1395) than the medium and small pore zeolites (MFI, $k_m$: 0.0956; CHA, $k_m$: 0.0130). These data suggest that the rate of ABS migration is faster in zeolites with larger pores as intuitively expected. The migration of ABS was especially lower for the CHA structure, who's mixed catalyst displayed a similar deactivation rate (0.0124) to VWTi + Silica (0.0158) even for high aluminum content CHA ($Si/Al_2 = 9$). This latter behavior seems likely to be attributable to the comparable diameter of a bisulfate ion (4.12 Å) to the CHA pore size (4 Å), thus hindering the migration of the ABS due to steric effects[30,31].

In summary, we present a mixed catalyst system of $V_2O_5$–$WO_3$/$TiO_2$ and Y zeolite for $NH_3$-SCR which is remarkably stable under the simulated exhaust gas condition including large amounts of $SO_2$. The mixed zeolite in close contact with the vanadia catalyst can effectively absorb condensed ammonium bisulfate from the catalyst surface, thereby preventing physical deactivation of the catalyst during SCR operation at low temperatures. Using this strategy, we achieve both high $NO_x$ removal rate and $SO_2$ resistance at low temperatures that have never been compatible so far. Because sulfur species is stored in zeolite as ammonium sulfate, not as metal sulfate, it is easily decomposed at low temperatures, making this system completely

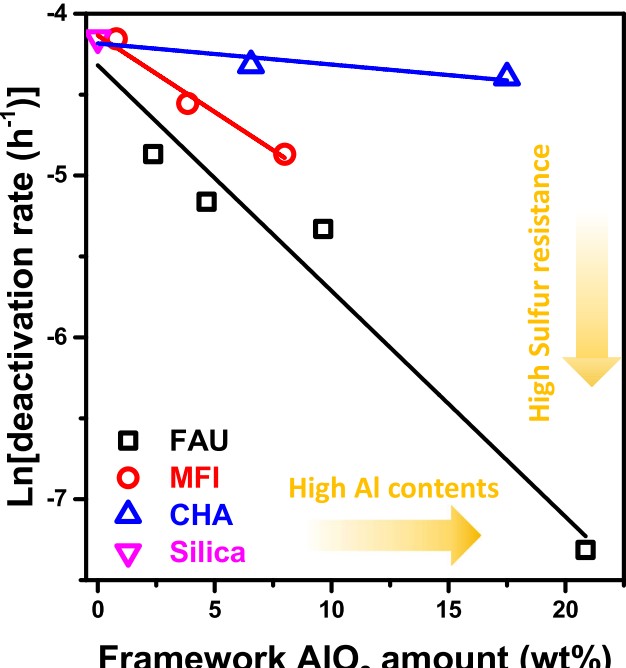

**Fig. 4 Mechanism investigations of ABS trapping with two determination factors.** The logarithm of the deactivation rate was plotted to the framework $AlO_2$ amount. Details of the reaction mechanism study are described in the reaction mechanism investigation section of the Supplementary Materials. Each data was obtained from $SO_2$ aging of physically mixed VWTi with Y zeolite (FAU, black), ZSM-5 (MFI, red), SSZ-13 (CHA, blue) and silica (magenta) (Supplementary Fig. 16). The amount of framework Al was calculated by using theoretical Al amount from $Si/Al_2$ ratio and $Al_{framework}/Al_{total}$ value from $^{27}Al$-NMR data (details in Supplementary Fig. 19). The linearity of the model was shown by linear regression of various zeolites (Supplementary Table 3). Data of silica was included in all fitting results for zero-point (at framework $AlO_2$ amount = 0).

reusable after regeneration. Reaction mechanism study demonstrates that the trapping ability of zeolites is dependent on the amount of framework Al and pore opening of structure that determine ABS migration rate. Our results suggest that simple physical mixing can be an effective solution to prevent catalyst deactivation. Because this strategy is not limited to vanadium catalyst, we expect that it can be utilized complementarily with other research on developing environmental SCR catalyst in the future.

## Methods

**Synthesis of catalyst and physical mixing with zeolite.** $V_2O_5$–$WO_3$–$TiO_2$ catalysts (VWTi) were synthesized by the wet impregnation method. Vanadium precursor solutions were prepared by adding ammonium metavanadate (Sigma Aldrich) dissolved in an oxalic acid solution (Sigma Aldrich). DT-52 $TiO_2$ support (WTi, 7.7 wt.% of tungsten, CRYSTAL) was added into the vanadium precursor solution and stirred for 30 min. The solution with DT-52 support was dehydrated by a using rotary evaporator and completely dried in a forced convection oven overnight at 105 °C. Dried catalysts were calcined at 500 °C for 4 h. The composition of the resulting VWTi material was measured by using ICP-AES (Supplementary Fig. 1c). The VWTi was mixed with various zeolites by grinding in a mortar. Y zeolite (FAU structure, $Si/Al_2$ = 5, 12, 30, 60, Alfa Aesar), ZSM-5 (MFI structure, $Si/Al_2$ = 23, 50, 250, Alfa Aesar) and SSZ-13 (CHA structure, $Si/Al_2$ = 9, 23, lab-made and provided from Heesung Catalysts, respectively) were used. The mixing ratio was 2:1 (VWTi: zeolite) in a mass ratio.

SSZ-13 ($Si/Al_2$ = 9) was synthesized by conventional hydrothermal method[32]. Briefly, 0.8 g of NaOH (Sigma Aldrich) and 25 g of $Na_2SiO_3$ (Sigma Aldrich) were dissolved in 52 mL of D.I. water. After vigorous stirring for 30 min under ambient conditions, 2.5 g of CBV 500 (Zeolyst) and 10.5 g of TMAdaOH (SACHEM) were

added and stirred for 30 min under ambient conditions. The prepared mixture was then transferred to 200 mL teflon-lined stainless steel autoclaves and placed in a forced convection oven at 140 °C. Ammonium ion exchange was repeated three times using 1 M ammonium nitrate solutions (Sigma Aldrich) at 65 °C to get an $NH_4^+$ form of SSZ-13. Cu-SSZ-13 was synthesized by using a conventional ion exchange method. 1 g of $NH_4^+$-SSZ-13 ($Si/Al_2 = 23$, provided from Heesung Catalysts) was added into 0.1 M copper nitrate precursor solutions (Sigma Aldrich), and stirred at 65 °C for 24 h. The mixture was then filtered and dried in a forced convection oven at 105 °C. Dried catalysts were calcined at 550 °C for 4 h. $Mn/TiO_2$ was synthesized by a using wet impregnation method. Manganese nitrate precursor solutions were prepared by adding 0.684 g of a manganese nitrate tetrahydrate (Sigma Aldrich) into 100 mL of D.I. water. 1 g of a microporous $TiO_2$ support (Lab made)[33] was added into the precursor solution and stirred for 30 min. The solution with microporous $TiO_2$ support was dehydrated by using a rotary evaporator and completely dried at 105 °C overnight. Dried catalysts were calcined at 500 °C for 4 h.

**Experimental tools for characterization of catalysts.** Micropore volumes of physically mixed catalysts were measured from Ar adsorption isotherms by using a Micromeritics 3Flex Surface and Catalyst Characterization instrument. Typically, 0.02–0.03 g of samples were loaded into the cell, and isotherms were obtained at −186 °C after degassing at 120 °C overnight. Micropore size distributions of samples were estimated from Horvath-Kawazoe method and plotted as differential pore volume curve (dV/dW vs. Pore width (W)). Solid-state $^{27}Al$-NMR spectra were obtained at 130.32 MHz on a Brucker Avance III HD (Brucker, German) under ambient condition (25 °C). All data were measured under magic angle spinning (MAS) at a spinning rate of 10 kHz. The pulse length was 2 μs, and the delay time was 0.1 s. TEM images were taken using a JEM-ARM2000F microscope at 200 kV with spherical aberration correction and cold FEG. Samples for TEM were prepared by drop-drying from their suspensions onto a 200 mesh carbon coated copper grid. Acetone was used as a solvent to prevent dissolution of ABS to the solvent as making suspensions of the sample. Line-EDS images were taken under a STEM mode. ICP-AES results were obtained by an OPTIMA 8300 (Perkin-Elmer) instrument to measure loading of vanadium and tungsten in VWTi. The amount of deposited sulfur was measured from Elemental Analysis by Flash2000 (Thermo Fisher Scientific). Mass spectrometry experiments were conducted on a HIDEN Analytical QGA using a secondary electron multiplier (SEM) detector. DRIFT spectra were obtained in a diffuse reflectance cell (Praying Mantis, Harrick) using a Fourier transform infrared (FT-IR) spectrometer (IS-50, Thermo Fisher Scientific). Outlet gas analyses were also performed using FT-IR spectroscopy (Nicolet6700, Thermo Fisher Scientific) with a 2 m gas cell (Gemini, International Crystal Laboratories). An MCT (Mercury-Cadmium-Telluride) type detector was used for all FT-IR analyses. Raman spectra were obtained on a Thermo DXR2xi with a 532 nm laser.

**Experimental setup of NH₃-SCR reaction system.** Reactivity and $SO_2$ poisoning data for the NH₃-SCR reaction were measured in a down-flow 1/4" ID tubular quartz reactor. All samples were pelletized and sieved to 180–250 μm particles to prevent pressure drop. Reactions were performed under 500 ppm NO (5000 ppm in $N_2$, Deokyang Co., Ltd.), 600 ppm NH₃ (5000 ppm in $N_2$, Deokyang Co., Ltd.), 10% $O_2$ (99.995%, Daesung industrial gases Co., Ltd.), 5% $CO_2$ (99.999%, KS gas Co., Ltd.), 10% $H_2O$ (deionized, introduced from PURELAB Chorus, ELGA), 30 ppm $SO_2$ (when used, 1000 ppm in $N_2$, Deokyang Co., Ltd.), and balance $N_2$ (99.999%, Daesung industrial gases Co., Ltd.). The gas hourly space velocity (GHSV) was 150,000 mL/h·$g_{cat}$. For the case of physical mixture samples, the GHSV was set based on a weight of VWTi. $NO_x$ concentrations were recorded using a $NO_x$ chemiluminescence analyzer (42i High level, Thermo Scientific), with $NO_x$ conversions calculated using the Eq. (2).

$$NO_x \text{ conversion } (\%) = \frac{[NO_x]_{in} - [NO_x]_{out}}{[NO_x]_{in}} \times 100 \quad (2)$$

NH₃-SCR activities of the catalysts were measured at 220 °C under NH₃-SCR reaction conditions. For testing resistance to $SO_2$ poisoning, the samples were aged for 22 h under NH₃-SCR conditions with 30 ppm or 100 ppm $SO_2$. After the $SO_2$ aging process, aged catalysts were regenerated at 350 °C for 2 h under 10% $O_2$, 5% $CO_2$, 10% $H_2O$, and balance $N_2$. The experimental protocol scheme is presented in the Supplementary Fig. 18. This protocol was repeated for multiple operations.

To obtain Arrhenius plots, the rate of $NO_x$ consumption was calculated, and $ln(-r_{NO_x})$ vs. 1/T was plotted using the Eq. (3). C values are the $NO_x$ concentrations measured by the $NO_x$ analyzer in ppm, $V_{total}$ is the total volumetric flow rate which is 0.2 L/min, P is 1 atm, T is ambient temperature, and R is the gas constant.

$$-r_{NO_x}\left(mol_{NO_x} s^{-1}\right) = \frac{\left(C_{NO_x,in} - C_{NO_x,out}\right)}{1000000} V_{total}\left(\frac{P}{RT}\right) \quad (3)$$

**Reaction mechanism investigations of ABS migration.** Reaction mechanism investigation of ABS trapping was performed by using two determination factors; the amount of Al and the zeolite structure. Some assumptions were made to simplify the model. First of all, the degree of deactivation is proportional to the deposited amount of ABS because the deactivation of VWTi is due to physical poisoning[19,21]. Therefore, the deactivation rate of the catalyst is assumed to be proportional to the deposition rate of ABS on the VWTi (Eq. 4). ABS deposition rates are composed of two terms: the rate of ABS formation on VWTi, and migration rates to the zeolite. The rate of ABS formation on VWTi is expected to be constant ($K_f$) because VWTi catalysts were almost linearly deactivated during 22 h (Fig. 1a). The migration rates of ABS are hypothesized to be proportional to the amount of ABS on VWTi ($C_{ABS}$) and framework Al ($C_{Al}$) in the zeolite. $k_m$ is the rate constant of ABS migration depending on the type of zeolite framework. At last, $C_{Al}$ is approximated to be the initial amount of Al ($C_{Al0}$), so ignoring the effect of Al blocking by ABS (Eq. 5). It seems quite reasonable because only 10% of total micropore volume was blocked by migrated ABS after $SO_2$ aging for 22 h, thereby implying less effect of accumulated ABS on $C_{Al}$ (Fig. 2e). Solving the differential equation reveals that D is proportional to exponential of $k_m$ and $C_{Al0}$ (Eq. 6). Therefore, the logarithm of the deactivation rate is proportional to the product of $k_m$ and $C_{Al0}$ (Eq. 7). Thus, $C_{Al0}$ and $k_m$ indicate the two determination factors suggested above; notably, Al amount and zeolite structure, respectively. From these proposed relationships, the logarithm of the deactivation rate was plotted versus the amount of framework alumina (Fig. 4), with the amount of framework alumina quantified from theoretical $Si/Al_2$ ratios and $^{27}Al$-NMR (Supplementary Fig. 19).

$$\text{Deactivation rate}(D) \propto \frac{dC_{ABS}}{dt} \, (C_{ABS} : concentration \, of \, ABS \, on \, VWTi) \quad (4)$$

$$\frac{dC_{ABS}}{dt} = r_{ABS\,formation} - r_{ABS\,migration} = K_f - k_m C_{ABS} C_{Al0} \quad (5)$$

$$D \propto \frac{dC_{ABS}}{dt} = K_f \cdot e^{-k_m C_{Al0} t} \quad (6)$$

$$ln(D) \propto k_m \cdot C_{Al0} \quad (7)$$

**Computational details.** The periodic density functional theory (DFT) calculations have been carried out using the Vienna ab initio simulation package (VASP)[34,35]. The core-valence interactions were treated by Blöchl's projector augmented wave (PAW) approach[36]. Perdew–Burke–Ernzerhof (PBE) functionals, based on the generalized gradient approximation (GGA), were used to account for exchange–correlation[37]. A Gaussian smearing method, with a width of 0.05 eV, was applied to determine the partial occupancies. The conjugate gradient algorithm was used for geometry relaxations until the forces on all of the unconstrained atoms were less than 0.03 eV Å⁻¹. For all calculations, spin polarized computations were performed and the energy cutoff of the plane-wave expansion was set to 500 eV.

For bulk anatase $TiO_2$, a $6 \times 6 \times 6$ Monkhorst–Pack k-point mesh was used[38]. The optimized lattice parameters for bulk $TiO_2$ were $a = 3.83$ Å, $b = 3.83$ Å, $c = 9.62$ Å, and $\alpha = \beta = \gamma = 90°$. The $TiO_2(001)$ surface was modeled with a $(4 \times 2)$ surface unit cell with a three layer thickness and 20 Å of vacuum between the slabs (Supplementary Fig. 20)[39]. The top two layers were allowed to relax, and the bottom layer was fixed[40]. For all slab calculations, a $2 \times 4 \times 1$ Monkhorst–Pack k-point mesh was used, and a dipole correction was included. To model our VWTi catalyst, the strong interactions between the support and the active phase were used. This model was constructed by placing half monolayer of oxygen atoms on top of a pseudomorphic $VO_2$ phase that extends over the (001) $TiO_2$ anatase structure. This created a vanadia monolayer that does not resemble the structure of bulk $V_2O_5$ but had the fully oxidized $V_2O_5$ stoichiometry. To study the effect of adding tungsten in our system, 50% of vanadia was replaced by $WO_3$ to obtain a monolayer of 50% $V_2O_5$-50% $WO_3$ for the active phase (Supplementary Fig. 20a). The vanadia and tungsten oxide species were placed on the top part of the $TiO_2(001)$ surface, and the stability of this system was checked by calculating the surface formation energy as following Eq. (8).

$$E_f = \frac{1}{A}\left[E_{VW/TiO_2(surf)} - E_{TiO_2(surf)} - E_{V_2O_5(bulk)} - E_{WO_3(bulk)}\right] \quad (8)$$

where $E_f$ is the surface formation energy, $E_{VW/TiO_2(surf)}$ is the total energy of the surface cell of the supported vanadia and tungsten oxide active phase on titania system, $E_{TiO_2(surf)}$ is the total energy of the $TiO_2(001)$ anatase surface cell, $E_{V_2O_5(bulk)}$ is the total energy of the $V_2O_5$ bulk unit cell, and $E_{WO_3(bulk)}$ is the total energy of the $WO_3$ bulk unit cell. The calculated surface formation energy at 0 K is found to be $-0.52$ J/m². 

Faujasite (FAU) zeolite was modeled by a periodic low-symmetry rhombohedral unit cell[41]. The unit cell contains 144 atoms; 48 Si and 96 O atoms. Four Si atoms were replaced by four Al atoms following the Löwenstein rule[42]. This provided a FAU zeolite model with a Si/Al ratio of 11 which is close to the FAU zeolite sample ($Si/Al_{framework} = \sim 9$) used in this study. The negative charge on the lattice was compensated by four protons introduced to the structure at O1 sites (Supplementary Fig. 20b). For the FAU zeolite model, the Brillouin zone sampling was restricted to the gamma point and the optimized lattice parameters were $a = b = c = 17.49$ Å, and $\alpha = \beta = \gamma = 60°$.

We considered the dual-site adsorption of ABS molecule on VWTi catalyst[43]. On VWTi catalyst surface, $NH_4^+$ and $HSO_4^-$ of ABS molecule are, respectively, adsorbed as $H_3N-H\cdots O-V/W$ and adjacent $HO_3S-O\cdots W$ bonds. On FAU zeolite, it is more likely that ABS molecules are adsorbed on the internal surface rather than the external surface. The stability of ammonium bisulfate (ABS) molecules on the VWTi catalyst surface and the internal surface of FAU zeolite was checked by calculating the adsorption energies of ABS molecules, water molecules, and hydrated ABS molecules according to the following Eq. (9).

$$E_{ads} = E_{adsorbent+adsorbate} - E_{adsorbent} - E_{adsorbate}, \qquad (9)$$

where $E_{ads}$ is the adsorption energy, $E_{adsorbent + adsorbate}$ is the total energy of the adsorbent with the adsorbed species, $E_{adsorbent}$ is the total energy of bare adsorbent, and $E_{adsorbate}$ is the total energy of the free adsorbates. Isolated adsorbates were calculated in a $20 \times 20 \times 20$ Å$^3$ periodic box. According to this definition, a larger negative $E_{ads}$ value indicates higher stability. Since the electrostatic interaction energy was very large, any starting double ions of $NH_4^+$ and $HSO_4^-$ for pure ABS molecules were collapsed into its neutral form[44]. This was mainly attributed to the fact that there were no water molecules to separate the two ions. Therefore, pure ABS molecules were not expected to be stable in the gas phase but rather were likely to exist as paired ions on the catalyst and FAU zeolite surfaces.

## Data availability

All data that support the findings of this study are available within the paper and its Supplementary Information or from the corresponding author upon reasonable request.

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

## Acknowledgements

We thank the Research Institute of Industrial Science and Technology (RIST) for funding as a research project (5261-20190003). This research was also supported by the Basic

Science Research Program through the National Research Foundation of Korea (NRF), funded by the Ministry of Science, ICT and Future Planning (MSIP) (NRF-2016R1A5A1009592). All data is available in the main text or the supplementary materials. D.H.K., I.S., H.L., S.W.J., J.T.L., J.K., Y.B. and D.J.K. are inventors on Korean patent application (1020190120527) submitted by Pohang Iron and Steel Company (Posco Co., Ltd.), RIST, and Seoul National University (SNU) R&DB Foundation. The authors thank Prof. T. Hyeon, Prof. J. Park, and Dr. C. H. F. Peden for very kindly reviewing the manuscript.

## Author contributions

I.S. and H.L. equally contributed to the work. I.S. and H.L. conceived the research ideas and designed the experiments, and S.W.J. contributed to the reaction measurements and data analysis. I.A.M.I. carried out the theoretical studies. J.W.H. supervised the theoretical calculations. J.K., Y.B., and D.J.K. commented on lab-scale data and provided information on industrial SCR operations. D.H.K. conceptualized and supervised the project. I.S. and H.L. co-wrote the manuscript under the supervision of D.H.K. All of the authors discussed the results and commented on the manuscript.

## Competing interests

The authors declare no competing interests.
