## [Peer Review File · Nature Communications]

REVIEWER COMMENTS

Reviewer #1 (Remarks to the Author):

In this work, the traditional vanadia catalyst were physically mixed with H-Y zeolite which can effectively trap liquid ABS to avoid physical deactivation by the deposition of ABS. Furthermore, the trapped ammonium sulfate salts were in an unstable condition, which could be easily decomposed at temperatures as low as 350 °C to regenerate the catalyst. These results were demonstrated by rational and logical experimental design, kinetic modelling and computational calculations. This work provides a novel SO₂-tolerance strategy through the absorptive protection which is meaningful to further develop SCR catalysts. However, there are some suggestions for this work :

1. As mentioned in the Introduction section, you think that the only feasible option for enhance the low-temperature activity of V-based catalysts is to increase the number of active V sites, which can accelerate the formation of ABS. Hence, the promotion of both the activity and SO₂-tolerance is a huge dilemma. However, in this work, through mixed with H-Y zeolite physically, vanadia catalysts did not obtain an enhanced low-temperature activity, which might be in consistent with your initial purpose of this work. Furthermore, the preparation cost of zeolites is relatively high for stationary source, which might be a defect of your strategy. Please supplement your understanding and consideration about these issues.
2. In this work, the concentration of SO₂ used in tests was 30 ppm which might be much lower than that used in other research in this field. And, according to the discussion around the amount of deposited sulfur species over the catalysts (Supplementary Table 1), ABS trapped by zeolite could accumulate continuously until the catalysts were regenerated at 350 °C. Hence, whether the adsorption capacity of zeolites can satisfy the condition with SO₂ in high concentration should be considered. Please supplement corresponding tests with SO₂ in a higher concentration (100~200 ppm) or supplement some convincing references and reports to support your option in SO₂ concentration.
3. In the discussion around the absorptive protection mechanism, the change of the slope in the Arrhenius plots of the ABS/VWTi+Z PM L material was be attributed to the physical deactivation of the phase transformation of ABS over the VWTi catalyst. However, if the deactivation is just caused by the physical behavior of ABS like the deposition covered the active sites, the activation energy should not change. So, please provide rational explanation of this phenomenon.
4. Meanwhile, you think that VWTi in ABS/VWTi+Z was not deactivated by ABS according to the linear Arrhenius plot. However, just mentioned above, the reduction of the number of active sites will not change the activation energy. Hence, Arrhenius plots might not suitable for understanding the influence of ABS over catalysts. Please supplement other evidence to prove your conclusion.
5. To verify the effects of residual aluminum sulfate on the ABS trapping ability of the zeolite, sulfur-saturated zeolites were investigated that the residual sulfur will not influence the overall reusability of the VWTi+Z system. Meanwhile, through the computational comparison of the affinity of the two materials for ABS on the VWTi surface and H-Y zeolite (Fig. 3b), you think that water molecules in the faujasite structure of H-Y zeolite enhanced the stability of ABS through the interaction with Brønsted acid sites generated by the framework Al, which rationalizes the migration of ABS between particles.

However, as shown in Supplementary Fig. 6, in the Sulfated Z, hydroxyl group from alumina decreased compared to fresh Y zeolite, which is contradictory to your deduction before. Please provide more precise and rational evidence and explanation.

6. In the part of DFT calculation (Fig. 3b and Supplementary Fig. 8), you have compared the adsorption energy of water, ABS, hydrated ABS + one water, and hydrated ABS + three water on the VWTi catalyst with those inside H-Y zeolite. And, through discussing the factors determining the ABS trapping ability, the pore size of zeolites were thought to be one of key factors, which might have steric effects to influence the migration of ABS. However, you have only calculated the adsorption energy of molecule mentioned above over the vanadium oxide and inside the zeolite and these species might be adsorbed over the surface of zeolite instead of the inside of framework. Please supplement the DFT calculation of the adsorption energy of water, ABS, hydrated ABS + one water, and hydrated ABS + three water on the surface of zeolite.

7. In the part of DFT calculation, the detailed information of elements corresponding to the spheres with different color should be illustrated in the legend (Fig. 3b, Supplementary Fig. 8 and Supplementary Fig. 13).

8. Most recently, great progress has been made to prevent sulfur deactivation of catalysts for NH₃-SCR of NO, e. g. Appl Catal B-Environ, 269 (2020) 118825; Environ Sci Technol, 53 (2019) 6462-6473. These related publications should be involved and summarized in the background of this work.

Reviewer #2 (Remarks to the Author):

This paper reported the prevention from sulfur deactivation of vanadia catalysts in NH₃-SCR. Some results are interesting. However, this study does not meet the level of this journal. I recommend publishing in the other journal after the major revision.

The other comments are below.

1. Concept of the regeneration in NH₃-SCR

In Page 3 line 33, 'incinerators, boilers, and power plants, where NH₃-SCR is equipped, have off-gas temperatures that are usually below 250 °C' is misleading. In the case of boilers, and power plants, the catalyst was placed just downstream of the boiler to get the high reaction temperature if the scale of the boiler is large. In the small boilers such as incinerators, the exhaust gas is filtered to remove particulate and then removed SO_x, and then de-NO_x. Only such case the off-gas was reheated. The system will depend on the scale of the boiler and fuel. The technic reported cannot apply to all of the NH₃-SCR systems.

And regeneration of the catalysts by heating has already been in practical use. The authors claim is adding adsorption materials into the catalyst. This idea does not meet the level of this journal because it will not solve the sulfur problems in a fundamental way.

2. The quantification of SO₂

The quantification of SO₂ should be checked in the reaction condition. The amount converted to NH₄HSO₄ depends on the oxidation activity of the catalysts from SO₂ to SO₃. If the catalyst is active, the deactivation will become quick. The speed of the deactivation should be explained from various perspectives.

3. The state of zeolite

In the reaction conditions, all of the acid sites should be neutralized by NH₃. Therefore, the DFT calculation has to take into consideration the existence of NH₄⁺. And the authors ignored the possibility that SO₃ reacts with the NH₄⁺ and water in the pore.

4. Regeneration of the catalysts

The analysis of the regeneration of the catalysts should be analyzed by TPO and TG in actual conditions for all catalysts. The authors described that the decomposition of NH₄HSO₄ needs 700 degrees C. However, the decomposition under inert gas flow and oxidative decomposition is different. The authors need to reveal the regeneration of the catalysts under 350 degrees C.

Reviewer #3 (Remarks to the Author):

This is a very strong and novel paper, where it is shown that the addition of zeolite to a vanadia catalyst can reduce the sulfur poisoning. I recommend that the paper is considered for publication after updating the paper according to the comments below:

- On line 121 it is proposed that liquid ABS is formed. This statement should be further backed-up with experiments, or in more detail explain which experimental results that shows this.
- The sulfur is released with a main peak around 415C, see Fig 3 supp. However, the regeneration is performed at 350C. How can the regeneration be performed at a lower temperature than the sulfur release? Also the mixed vanadia/zeolite catalyst exhibit SO₂ desorption at 415 and 715C. There is a risk that the catalyst would be saturated with sulfur which would result in a degradation and that it is no longer possible to regenerate the catalyst at 350C. An experiment where the catalyst is saturated with sulfur, could for example be done by repeated cycles with much higher SO₂ concentration in order to examine that the regeneration still is functioning.
- The kinetic model is more simplified simulations for investigating a reaction mechanism. A kinetic model should include the effect of temperature. It is also assumed in the model that the rate is not depending on the SO₂ concentration, which is quite probable. If it should be stated that a kinetic model is performed it must be done in much more detail. Alternatively, do not claim that it is a kinetic model but instead reaction mechanism investigations.
- Figure 2: Use real distance on x-axis instead of arbitrary units.
- Clarify briefly in the text in the results section regarding concentration during poisoning and regeneration to facilitate the reading.

Response to the reviewers' comments

Sincerest thanks for your response and referees' valuable advice and critical comments on our manuscript. To fully address the reviewers' comments, the additional experiments were conducted and manuscript was carefully revised. Details are shown below, including a point by point response to the reviewers' comments marked in blue and newly added sentences in the revised manuscript marked in red, and we hope that the revisions will comply with the referees' remarks. We hope that a revised version of the manuscript will be processed in due course by *Nature Communications*.

[Reviewer #1]

In this work, the traditional vanadia catalyst were physically mixed with H-Y zeolite which can effectively trap liquid ABS to avoid physical deactivation by the deposition of ABS. Furthermore, the trapped ammonium sulfate salts were in an unstable condition, which could be easily decomposed at temperatures as low as 350 °C to regenerate the catalyst. These results were demonstrated by rational and logical experimental design, kinetic modelling and computational calculations. This work provides a novel SO₂-tolerance strategy through the absorptive protection which is meaningful to further develop SCR catalysts. However, there are some suggestions for this work.:

Question 1. As mentioned in the Introduction section, you think that the only feasible option for enhance the low-temperature activity of V-based catalysts is to increase the number of active V sites, which can accelerate the formation of ABS. Hence, the promotion of both the activity and SO₂-tolerance is a huge dilemma. However, in this work, through mixed with H-Y zeolite physically, vanadia catalysts did not obtain an enhanced low-temperature activity, which might be in consistent with your initial purpose of this work. Furthermore, the preparation cost of zeolites is relatively high for stationary source, which might be a defect of your strategy. Please supplement your understanding and consideration about these issues.

Answer: Thank you for the valuable comment. First, the conventional VTiO₂ catalysts have contained 2~3 wt.% of vanadia, which show great NO_x removal ability at conventional operating temperatures (300~400 °C). Recently, industrial fields have required the low temperature NO_x removal system (< 250 °C) for high fuel efficiency, strengthened NO_x regulation and stability of the system, however, conventional VTiO₂ catalysts have poor NO_x conversion at the low temperature. As we mentioned in the manuscript, increasing the vanadia loading is the most feasible way to improve the NO_x removal ability of conventional catalysts. And also mentioned in manuscript, there is deactivating issue in the high loading vanadia/titania catalysts due to the ABS formed from SO₂ contained in fuel. Here, the dilemma arises. As shown in Fig. R1, conventional catalyst with 2 wt.% vanadia (2VWTi) showed great sulfur resistance during 22 h SO₂ aging, however, its NO_x removal ability at 220 °C was about 24 % of NO_x conversion, which was insufficient to be utilized in low-

temperature SCR system. To enhance the low-temperature activity of vanadia/titania catalyst, we increased the content of vanadia from 2 wt.% to 5 wt.%, which gave rise to improvement of NO_x conversion to 65%. In spite of excellent low temperature activity of 5wt% vanadia/titania (VWTi), it was deactivated under SO₂ contained reaction condition from 65% to 45% in the NO_x conversion due to the deposition of ABS as it was described in our manuscript. Adding the promoters is also one of the way to enhance the activity, however, it gives rise to same result because those promoters also enhance the SO₂ oxidation, which is known as the elementary reaction of ABS formation. Our new strategy proposed here is to resolve such dilemma; to improve the low temperature NO_x removal ability, we increase the loading of vanadia to 5wt. %, which is more than twice the amount contained in conventional catalyst. And to solve the followed SO₂ deactivation issue, we introduced the ‘physical mixed zeolite’ that carried out the adsorption of deactivating material (ABS).

Fig. R1. Comparison of 22 h SO₂ aging profiles of VWTi (5 wt.% of Vanadia), VWTi+Y with 2VWTi (2 wt.% of Vanadia) that is conventional catalyst under NH₃-SCR condition with SO₂ at 220 °C (500 ppm NO, 600 ppm NH₃, 10% O₂, 5% CO₂, 10% H₂O, 30 ppm SO₂ and balance N₂, GHSV : 150,000 mL/h·g_{cat} based on VWTi weight).

Secondly, we also agree with high cost of the zeolite. However, Y zeolite that is used in this study is relatively cost-efficient than other zeolites. Furthermore, although the ratio of VWTi catalyst to the zeolite is 2:1 in this study, much less amount of zeolite could be effective to improve the SO₂ resistance of the catalysts, which will make the system more economical to implement in stationary sources. For example, when we lowered the ratio of VWTi:Zeolite to 16:1 (Fig. R2), the deactivation rate was quite comparable to that of the 2:1 mixture. In addition, the proposed catalyst has been successfully implemented and employed in the sintering furnace of the POSCO steel manufacturing company.

Fig. R2. 22 h SO₂ aging profiles of VWTi+Y with different ratio of mixture under NH₃-SCR condition with SO₂ at 220 °C (500 ppm NO, 600 ppm NH₃ 10% O₂, 5% CO₂, 10% H₂O, 30 ppm SO₂ and balance N₂, GHSV : 150,000 mL/h·g_{cat} based on VWTi weight).

Revised manuscript (page 6, line 109): The VWTi catalyst with high V loading (containing 5 wt.% V₂O₅) showed a moderate activity of ~65% conversion, and this is a fairly good low temperature performance compared to conventional catalyst with 2 wt.% V₂O₅ loading, which only exhibits ~24% conversion (not shown).

Question 2. In this work, the concentration of SO₂ used in tests was 30 ppm which might be much lower than that used in other research in this field. And, according to the discussion around the amount of deposited sulfur species over the catalysts (Supplementary Table 1), ABS trapped by zeolite could accumulate continuously until the catalysts were regenerated at 350 °C. Hence, whether the adsorption capacity of zeolites can satisfy the condition with SO₂ in high concentration should be considered. Please supplement corresponding tests with SO₂ in a higher concentration (100~200 ppm) or supplement some convincing references and reports to support your option in SO₂ concentration.

Answer: Thank you for the suggestion of additional experiments. We performed the SO₂ aging protocol for the VWTi+Y catalyst under NH₃-SCR condition with 100 ppm SO₂ (Supplementary Fig. 4). Even under harsher condition, the catalyst demonstrated excellent SO₂ resistance. This result indicates that the adsorption capacity of the Y zeolite (Si/Al₂=12) is sufficient to be utilized in the industry.

Supplementary Fig. 4.

Reaction test under SO₂ in higher concentration.

Reaction profiles of 22 h SO₂ aging over VWTi+Y under NH₃-SCR condition with different SO₂ concentration at 220 °C (500 ppm NO, 600 ppm NH₃ 10% O₂, 5% CO₂, 10% H₂O, 30 ppm or 100 ppm SO₂ and balanced with N₂, GHSV : 150,000 mL/h·g_{cat} based on VWTi weight).

Revised manuscript (page 6, line 117): It was also confirmed that H-Y zeolite can prevent ABS deactivation even with much higher SO₂ concentration of 100 ppm (Supplementary Fig. 4).

Question 3 and 4. In the discussion around the absorptive protection mechanism, the change of the slope in the Arrhenius plots of the ABS/VWTi+Z PM L material was attributed to the physical deactivation of the phase transformation of ABS over the VWTi catalyst. However, if the deactivation is just caused by the physical behavior of ABS like the deposition covered the active sites, the activation energy should not change. So, please provide rational explanation of this phenomenon. Meanwhile, you think that VWTi in ABS/VWTi+Z was not deactivated by ABS according to the linear Arrhenius plot. However, just mentioned above, the reduction of the number of active sites will not change the activation energy. Hence, Arrhenius plots might not suitable for understanding the influence of ABS over catalysts. Please supplement other evidence to prove your conclusion.

Answer: Thank you for the valuable comment. We entirely agree with your suggestion because the poisoning by ABS is physical deactivation blocking the active sites, not the chemical deactivation. In physical deactivation, the activation energy should not change, and only pre-exponential factor (A) changes as you pointed out, which was also reported previously (C. Li et al., *Phys. Chem. Chem. Phys.* **19**, 15194-15206 (2017)). We conducted same experiments with 2 wt.% of ABS impregnated VWTi that is much lower amount, and Arrhenius plot is displayed (Supplementary Fig. 11). Above the melting point of ABS (160 °C), VWTi+Z and VWTi+Z PM L show same slopes of Arrhenius plots (61.1 kJ/mol and 58.3 kJ/mol, respectively), instead the pre-exponential factor decreased in the PM L sample ($1.8 \times 10^6 \rightarrow 6.7 \times 10^5$). This result clearly illustrates that the ABS gives rise to physical deactivation in the PM L catalyst. In the case of our data (Fig. 3a) in the manuscript, the amount of ABS was too much (10 wt.%) to observe the same result, because migration of ABS might be still on-going at 160~215 °C. However, the data still reveal the importance of physical contact between VWTi and zeolite particles in ABS trapping.

Supplementary Fig. 11.

a, Arrhenius plots of 2 wt.% ABS/VWTi+Z (black dot) and 2 wt.% ABS/VWTi+Z PM L (red dot) during transient NH_3 -SCR reaction (temperature from 100 °C to 220 °C with ramp rate 1 °C/min, other reaction conditions were the same as steady-state NH_3 -SCR experiment. The 2 wt.% ABS/VWTi sample was prepared by wet impregnation method of 2 wt.% of ABS on the VWTi catalyst. **b**, Activation energy (E_a) and pre-exponential factor obtained from the

Arrhenius plots of 2wt ABS/VWTi+Z and 2wt ABS/VWTi+Z PM L during the transient SCR reaction.

Revised manuscript (page 9, line 197): We conducted same experiments with much lower amount of pre-impregnated ABS (2 wt.%) on VWTi (Supplementary Fig. 11). Above the melting point of ABS (160 °C), VWTi+Z and VWTi+Z PM L show same slopes of Arrhenius plots while only pre-exponential factor decreased in the PM L sample, which clearly illustrates that the ABS gives rise to physical deactivation in the PM L catalyst.

Question 5. To verify the effects of residual aluminum sulfate on the ABS trapping ability of the zeolite, sulfur-saturated zeolites were investigated that the residual sulfur will not influence the overall reusability of the VWTi+Z system. Meanwhile, through the computational comparison of the affinity of the two materials for ABS on the VWTi surface and H-Y zeolite (Fig. 3b), you think that water molecules in the faujasite structure of H-Y zeolite enhanced the stability of ABS through the interaction with Brønsted acid sites generated by the framework Al, which rationalizes the migration of ABS between particles. However, as shown in Supplementary Fig. 6, in the Sulfated Z, hydroxyl group from alumina decreased compared to fresh Y zeolite, which is contradictory to your deduction before. Please provide more precise and rational evidence and explanation.

Answer: Thank you for the important comments. To investigate the Al sites of Y zeolite in detail after sulfation, we performed ²⁷Al-solid NMR experiments over the Y zeolite and sulfated Y zeolite (Supplementary Fig. 14a). Y zeolite shows two resonance peaks at 55 ppm and 0 ppm that are assigned to framework Al with tetrahedral coordination and extra-framework Al with octahedral coordination, respectively. After sulfur saturation, the framework Al peak dramatically decreases and octahedral coordinated Al increases with a little peak shift. New peak appears at -25 ppm, which cannot be assigned yet. From the ²⁷Al-NMR, we can conclude that the Al sites in the Y zeolite form strong interaction with sulfate, especially the framework Al that acts as Brønsted acid site.

In addition, NH₃-TPD was conducted to investigate the acid properties of Y zeolite and sulfated Y zeolite (Supplementary Fig. 14b). There are two desorption peaks at 125 °C and 305 °C, which are assigned to desorbed NH₃ on Lewis acid sites and Brønsted acid sites, respectively. Although there is a huge decrease in the resonance peak of framework Al after the sulfation, little change is observed in the amount of Brønsted acid NH₃ on the S saturated Y zeolite, instead the amount of Lewis acid sites slightly decreases. This result clearly demonstrates that the amount of Brønsted acid sites, which play a significant role in ABS migration, is unchanged by the interaction with sulfate, because the sulfated sites also act as the Brønsted acid sites, which is well known chemistry in sulfated metal oxide (S. Yang et al., *Appl. Catal. B* **5**, 19-28 (2013)).

Supplementary Fig. 14.

Characteristics of Y zeolite and Sulfated Y zeolite.

a, ²⁷Al-NMR spectra of Y zeolite and Sulfated Y zeolite. **b**, NH₃-TPD profiles of Y zeolite and Sulfated Y zeolite. NH₃ adsorbed at 50 °C for NH₃ saturation and purged under He. Then, the samples were ramped to 500 °C with the rate of 5 °C/min.

Revised manuscript (page 11, line 229): One question is how sulfur-saturated zeolite can trap ABS, as observed above, where the framework Al has a strong interaction with sulfate groups. We observed that the amount of Lewis acid sites decreases slightly after sulfation, but little change in the amount of Brønsted acid sites (Supplementary Fig. 14). This result clearly demonstrates that the amount of Brønsted acid sites that play an important role in ABS migration is not changed by interactions with sulfate groups. It might be because the sulfated sites also act as the Brønsted acid sites, a well-known chemistry in sulfated metal oxides²⁷.

Question 6. In the part of DFT calculation (Fig. 3b and Supplementary Fig. 8), you have compared the adsorption energy of water, ABS, hydrated ABS + one water, and hydrated ABS + three water on the VWTi catalyst with those inside H-Y zeolite. And, through discussing the factors determining the ABS trapping ability, the pore size of zeolites were thought to be one of key factors, which might have steric effects to influence the migration of ABS. However, you have only calculated the adsorption energy of molecule mentioned above over the vanadium oxide and inside the zeolite and these species might be adsorbed over the surface of zeolite instead of the inside of framework. Please supplement the DFT calculation of the adsorption energy of water, ABS, hydrated ABS + one water, and hydrated ABS + three water on the surface of zeolite.

Answer: We highly appreciate the reviewer's comment. In fact, due to the smaller size of ABS species than the pore size of zeolites, it is more likely that ABS species are adsorbed on the internal surface rather than the external surface of zeolites. This was confirmed by our Ar adsorption-desorption results through blocking of the micropores. Although diffusion of large and branched hydrocarbon molecules into zeolite pores may be limited by the pore size, it is entirely appropriate to focus on the internal cavities in situations where the molecules of interest are easily accommodated in the zeolite pores.

Fig. R3. Adsorption of different base molecules at a Brønsted acid site in the internal cavity of FAU zeolite [Liu et al., *J. Phys. Chem. C* **121**, 23520–23530 (2017)].

Computationally, the external surface can be modeled by inserting a vacuum layer over the periodic repetitions of zeolites framework to eliminate the periodicity in a certain direction. However, at the external surface an obtained adsorption strength is only a pre-requisite for a reaction, due to the absence of the confining effect of the cavity which can provide a higher stability for the adsorbed molecules (T. Bucko et al., *J. Chem. Phys.* **117**, 7295, (2002); C. E. Hernandez-Tamargo et al., *J. Phys. Chem. C* **120**, 19097–19106 (2016)). Therefore, in our calculations, we only considered the adsorption of ABS species over the internal surface of FAU zeolites as described in the “Computational Details” part of the revised manuscript (F.

Schußler et al., *J. Phys. Chem. C* **115**, 21763–21776 (2011); C. Liu et al., *ACS Catal.* **5**, 7024–7033 (2015); C. Liu et al., *J. Catal.* **344**, 570–577 (2016); E. A. Pidko, *ACS Catal.* **7**, 4230–4234 (2017); C. Liu et al., *J. Phys. Chem. C* **121**, 23520–23530 (2017)). On VWTi catalyst surface, NH_4^+ and HSO_4^- of ABS molecule are respectively adsorbed as $H_3N-H\cdots O-V/W$ and adjacent $HO_3S-O\cdots W$ bonds (H. H. Phil et al., *Appl. Catal. B* **78**, 301–308 (2008)).

Revised manuscript (page 21, line 435): On FAU zeolite, it is more likely that ABS molecules are adsorbed on the internal surface rather than the external surface.

Question 7. In the part of DFT calculation, the detailed information of elements corresponding to the spheres with different color should be illustrated in the legend (Fig. 3b, Supplementary Fig. 8 and Supplementary Fig. 13).

Answer: Thank you for your comment. We add the illustration about the elements in the legend of figure regarding the DFT calculation.

Revised manuscript: Fig. 3, Supplementary Fig. 12 and 19.

Fig. 3. ABS trapping model by liquid phase migration. **a**, Arrhenius plots of ABS/VWTi+Z (black dot) and ABS/VWTi+Z PM L (red dot) during transient NH₃-SCR reaction (temperature from 100 °C to 220 °C with ramp rate 1 °C/min, **Reaction gas contains 500 ppm NO, 500 ppm NH₃, 10% O₂, 5% CO₂, and 10% H₂O balanced with N₂**; NO_x conversion profiles in Supplementary Fig. 7a). **b**, DFT calculation results regarding the stability of hydrated ABS by two H₂O molecules on the VWTi and H-Y zeolite. **c**, Schematic illustration of an operating principle of the ABS trapping in the VWTi+Z system.

Supplementary Fig. 12.

Optimized configurations used in DFT calculation.

a, Water adsorption energy on the VWTi catalyst and H-Y zeolite was calculated. **b**, Optimized configurations of ABS molecule without water on the VWTi and H-Y zeolite. ABS molecule is strongly bounded to catalyst surface through $\text{HO}_3\text{S}-\text{O}\cdots\text{W}$ chemical bond on the VWTi. In the case of H-Y zeolite, ABS molecule is stabilized through the bonding with Brønsted acid sites. **c**, Optimized configurations for hydrated ABS with one water molecule. **d**, Optimized configurations for hydrated ABS with three water molecules.

Supplementary Fig. 19.

a, Schematic representation of loading of vanadia and tungsten oxide monolayer on TiO₂ support. **b**, Schematic representation of constructing of FAU zeolite model with Si/Al ratio of 11.

Question 8. Most recently, great progress has been made to prevent sulfur deactivation of catalysts for NH₃-SCR of NO, e. g. Appl Catal B-Environ, 269 (2020) 118825; Environ Sci Technol, 53 (2019) 6462-6473. These related publications should be involved and summarized in the background of this work.

Answer: Thank you for the recommendation. We revised our introduction part adding the recent publication you suggested.

Revised manuscript (page 4, line 61): Recently, much effort has been made to solve the problem of sulfur deactivation in the SCR catalysts, for example, R. Yu *et al.* developed Cu-SSZ-13 zeolite-metal oxide hybrid catalyst that shows enhanced SO₂ tolerance by preferentially forming Zn sulfate over Cu sulfate (R. Yu *et al.*, *Appl. Catal. B* **269**, 118825

(2020)), and L. Han *et al.* discovered that a mesoporous TiO₂ shell can improve the SO₂ resistance of Fe₂O₃ catalyst (L. Han et al., *Environ. Sci. Technol.* **53**, 6462-6473 (2019)). Unfortunately, however, it is difficult to commercialize in the field because the method of preparing catalyst is complicated and a very high temperature (650 °C) is required to regenerate the deactivated catalysts.

[Reviewer #2]

This paper reported the prevention from sulfur deactivation of vanadia catalysts in NH₃-SCR. Some results are interesting. However, this study does not meet the level of this journal. I recommend publishing in the other journal after the major revision.

Question 1. (Concept of the regeneration in NH₃-SCR)

In Page 3 line 33, ‘incinerators, boilers, and power plants, where NH₃-SCR is equipped, have off-gas temperatures that are usually below 250 °C’ is misleading. In the case of boilers, and power plants, the catalyst was placed just downstream of the boiler to get the high reaction temperature if the scale of the boiler is large. In the small boilers such as incinerators, the exhaust gas is filtered to remove particulate and then removed SO_x, and then de-NO_x. Only such case the off-gas was reheated. The system will depend on the scale of the boiler and fuel. The technic reported cannot apply to all of the NH₃-SCR systems.

And regeneration of the catalysts by heating has already been in practical use. The authors claim is adding adsorption materials into the catalyst. This idea does not meet the level of this journal because it will not solve the sulfur problems in a fundamental way.

Answer: Thank you for the important comment. As reviewer pointed out, our work just targeted tail-end type SCR configuration that requires reheating to operate SCR catalyst. When the SCR catalyst was placed just downstream of boiler, i.e. high-dust configuration, the formation of ammonium bisulfate (ABS) is not an important problem, but flue gas contains other particulates including fly ash, arsenic, calcium, and other abrasive solids, and pore plugging or erosion of catalyst should be additionally considered (J. J. Schreifels et al., *Front. Energy* **6(1)**, 98-105 (2012)). Both SCR configurations have advantages and disadvantages, so they are currently in use, but it is expected that the ABS issues will become increasingly important due to the following reasons.

Unlike Europe and US, where environmental regulations have already been applied, in order for developing countries in Asia to cope with the strengthened environmental regulations, it is necessary to newly install SCR system in existing boilers with limited space (J. J. Schreifels et al., *Front. Energy* **6(1)**, 98-105 (2012); L. Muzio et al., *Fuel* **206**, 180-189 (2017)). In this case, tail-end SCR configuration is more preferred in retrofit installations due to SCR space requirements. Also, tens of ppm of SO₂ is present in flue gas even in the tail-end position when the fuel with high-sulfur contents are used such as power plant or steel-making furnace, which makes ABS-resistant catalyst essential.

In the case of incinerators burning biomass or solid waste, it is difficult to apply high-dust SCR configuration because various poisoning substances such as alkali, alkali metal, phosphorous or heavy metal are contained in flue gas, which can cause irreversible deactivation of catalyst. Therefore, tail-end SCR configuration is suitable to guarantee the lifetime of catalyst for these applications.

Furthermore, as the regulation of NO_x in the emission gas becomes more stringent, there has been an increased focus on operation of SCR at lower loads. It has been reported that a major change in the operation of the large coal-fired boilers. In the past, large coal-fired

power plants basically run at full-load, with few instances of lower load operation. However, these large coal-fired units are operating at lower loads recently, maybe with the increase in power generation from renewable and natural gas sources (L. Muzio et al., *Fuel* **206**, 180-189 (2017)). The temperature of flue gas entering the SCR catalyst can be decreased during these lower load operations, which can lead to potential ABS formation at the rear end of catalyst (Y. Shi et al., *Fuel Process. Technol.* **150**, 141-147 (2016); M. Zyrkowski et al., *E3S Web of Conferences* **137** 01021 (2019)). Thus, the importance of ABS-resistant catalyst is becoming more prominent considering that the regulation will be strengthened and the proportion of renewable energy will increase in power generation. In conclusion, as environmental regulations are strengthened and applied to various fields in the future, the applications of ABS-resistant catalysts will be diversified.

Meanwhile, the reviewer commented that adding SO₂ adsorption materials does not fundamentally solve the sulfur-poisoning problem. However, the amount of sulfur adsorbed on Y zeolite was insignificant when the pure Y zeolite was exposed to SO₂ under SCR reaction gas (Supplementary Fig. 3a), so it can be seen that the Y zeolite itself cannot act as SO₂ adsorbent under the reaction conditions. As described in our manuscript, Y zeolite plays a role in absorbing ammonium bisulfate produced from SO₃, not SO₂, only when mixed with vanadia catalyst. This is a unique advantage of zeolites, unlike reported SO₂ absorbing metal oxides such as Fe₂O₃ and CeO₂. Since the concentration of SO₃ produced by SO₂ oxidation at low temperature is much lower than that of SO₂, mixed zeolite can withstand a much longer time than metal oxides.

Fe₂O₃ or CeO₂ cannot be a fundamental solution, as the reviewer pointed out, because iron sulfate or cerium sulfate require a high temperature of 700 °C to decompose. However, sulfate trapped in zeolite can decompose at 350 °C since it is coordinated with ammonium ion rather than metal ion. That is why zeolite can function as an absorbing material capable of low-temperature regeneration. The developed catalyst has been implemented in an SCR system for the commercial sintering furnace in steel manufacturing process (POSCO Co., Ltd). The company expects that annual fuel costs savings will be ~8 million U.S. dollars, and the emissions of carbon dioxide will be reduced by ~30,000 tons per year, which is attributed to decrease in operating temperature.

Revised manuscript (page 3, line 46): However, as environmental regulations become more stringent and applied to various fields in the future, the importance of operating SCR at low-temperature below 250 °C becomes more important. For tail-end configurations in which the SCR reactor is placed downstream of precipitator or particulate control unit, the exhaust gas temperatures are usually below 200 °C, where the catalyst exhibits a much lower efficiency.

Question 2. (The quantification of SO₂)

The quantification of SO₂ should be checked in the reaction condition. The amount converted to NH₄HSO₄ depends on the oxidation activity of the catalysts from SO₂ to SO₃. If the

catalyst is active, the deactivation will become quick. The speed of the deactivation should be explained from various perspectives.

Answer: Thank you for your valuable comment. Unfortunately, our experimental setup cannot monitor the concentration of SO₂ because (i) SO₂ and SO₃ exists as an aerosol-like species in the presence of H₂O at 220 °C, making it difficult to detect with FT-IR gas cell, and (ii) long-term exposure to SO₂ and H₂O can corrode IR mirror and windows. However, the amount of deposited sulfur on catalyst after SO₂ aging experiment was measured and compared with the total amount of SO₂ injected.

Table R1. The quantification results of sulfur on various mixed catalysts after aging.

Catalyst	Deposited S on catalyst (μmol)	Injected SO ₂ on catalyst for 22 h aging (μmol)
VWTi	9.5	324.2
VWTi+Silica	9.5	
VWTi+Y (Si:Al ₂ :5)	13.3	
VWTi+Y (Si:Al ₂ :12)	11.3	
VWTi+Y (Si:Al ₂ :60)	9.3	

As shown in Table R1, about 3-4 % of injected SO₂ remained on VWTi and VWTi+Y catalyst after aging experiment. Considering that the amount of NH₄HSO₄ depends on the amount of SO₃ produced, it can be speculated that the conversion rate of SO₂ to SO₃ for VWTi is about 3 % under the reaction condition. This value is in good agreement with the reported previous work dealing with SO₂ oxidation on V₂O₅-WO₃/TiO₂ catalysts (J. P. Dunn et al., *Appl. Catal. B* **19**, 103-117 (1998)). We also found that the amount of sulfur on the VWTi+Y zeolite catalyst gradually increased as the Si:Al₂ ratio decreased. As the reviewer pointed out, this is because the higher the Al content, the better the ABS sorption ability, so that the activity of VWTi catalyst can be maintained without deactivation during SO₂ aging experiment.

Question 3. (The state of zeolite)

In the reaction conditions, all of the acid sites should be neutralized by NH₃. Therefore, the DFT calculation has to take into consideration the existence of NH₄⁺. And the authors ignored the possibility that SO₃ reacts with the NH₄⁺ and water in the pore.

Answer: Thank you for the very important comment. As reviewer suggested, we considered the possibility that gaseous SO₃ directly reacts with NH₄⁺ and H₂O in the pore of zeolite. In Supplementary Fig. 10b, SO₂ deactivation behavior of physically mixed VWTi+Z catalyst was compared with that of loose-contacted ‘VWTi+Z PM L’. The physically contacted area of VWTi and zeolite is very low in ‘VWTi+Z PM L’ catalyst, because VWTi and zeolite are individually sieved into particles and then mixed. It can be seen that ‘VWTi+Z PM L’ catalyst showed a similar deactivation rate to that of pure VWTi, which means that gaseous SO₃

cannot be captured directly by zeolite. Also in EDS analysis of SO₂-aged 'VWTi+Z PM L' catalyst, sulfur species are dominantly present on vanadia particle, not on zeolite (Fig. R4). The effect of alleviating sulfur deactivation is observed only in a well-mixed physical mixture, because the contact between VWTi and zeolite is necessary to trap liquid ammonium bisulfate.

Supplementary Fig. 10.

Comparison of physically mixed VWTi and Y zeolite (VWTi+Z) to loose contact sample (VWTi+Z PM L).

b, 22 h SO₂ aging profiles of VWTi, VWTi+Z, and VWTi+Z PM L under NH₃-SCR reaction condition with 30 ppm SO₂ at 220 °C. NH₃-SCR condition was under 500 ppm NO, 600 ppm NH₃, 10% O₂, 5% CO₂, 10% H₂O and balance N₂. GHSV: 150,000 mL/h·g_{cat} based on VWTi weight.

Fig. R4. TEM image and line-EDS of VWTi+Z PM L catalyst after SO₂ aging.

Meanwhile, we agree that the acid sites of catalysts can be initially neutralized by the basic ammonia molecules during SCR reaction. Therefore, we have additionally checked the stability of ammonia molecule and ABS with one ammonia molecule on VWTi and FAU zeolite (Supplementary Fig. 13). In the presence of ammonia on acid sites, the stability of ABS on FAU zeolite (-1.79 eV) was also higher than that on VWTi catalyst (-1.41 eV). The calculated results demonstrate that NH₃ and H₂O that present in the pores of the zeolite can stabilize ABS molecule.

Supplementary Fig. 13.

Optimized configurations used in DFT calculation.

a-b, Optimized configurations of ammonia adsorbed on the VWTi catalyst and H-Y zeolite. **c-d**, Optimized configurations of ABS molecule with pre-adsorbed ammonia on the VWTi catalyst and H-Y zeolite. **e**, Comparison of the calculated adsorption energy.

Revised manuscript (page 10, line 206): Also, SO₂ deactivation behavior of physically mixed VWTi+Z catalyst was compared with that of loose-contacted VWTi+Z PM L sample (Supplementary Fig. 10b). It can be seen that VWTi+Z PM L sample shows a similar deactivation rate to that of pure VWTi, demonstrating that gaseous SO₃ cannot be captured directly by zeolite. This result suggests that mixed H-Y zeolite could not work to alleviate sulfur deactivation without close physical contact to VWTi, likely because the H-Y zeolite traps sulfur via liquid phase ABS, not by gas phase SO₂ or SO₃.

Revised manuscript (page 11, line 222): Moreover, in presence of ammonia, the stability of ABS on H-Y zeolite was also much higher than that on the VWTi surface (Supplementary Fig.

13).

Question 4. (Regeneration of the catalysts)

The analysis of the regeneration of the catalysts should be analyzed by TPO and TG in actual conditions for all catalysts. The authors described that the decomposition of NH_4HSO_4 needs 700 degrees C. However, the decomposition under inert gas flow and oxidative decomposition is different. The authors need to reveal the regeneration of the catalysts under 350 degrees C.

Answer: We highly appreciate the reviewer's comment. As reviewer pointed out, the authors also understand that decomposition of ABS is largely different under inert gas and oxidative gas. According to our observations, sulfur species desorbed in the form of SO_2 under N_2 , but they desorbed in the form of various aerosol including sulfate ion (SO_4^{2-}) under O_2/N_2 flow. In the latter case, it is difficult to detect and quantify the sulfur species with FT-IR gas cell, and this is why we plotted thermal decomposition profiles of S-aged catalyst under inert flow (N_2) to clearly separate ammonium bisulfate and aluminum sulfate from the TPD data (Fig. 2f).

Fig. 2f. Thermal decomposition profiles of sulfur species on the SO_2 -aged VWTi+Z and VWTi catalysts.

In Fig. 2f, the peak at 440 °C originates from ammonium bisulfate, while the peak at 715 °C originates from aluminum sulfate. Under real regeneration conditions (H_2O , O_2 , CO_2/N_2), however, ammonium bisulfate species decompose much more easily than under N_2 conditions, so the catalyst can be regenerated at 350 °C, which will be described in the following.

Supplementary Fig. 6.

Outlet gas profiles of VWTi catalyst during regeneration step.

a, Ammonia and sulfate profiles during regeneration step which were obtained by using FT-IR with a gas cell. Before regeneration, VWTi catalyst was aged under SCR reaction gas containing 100 ppm SO₂ at 220 °C for 22 h. Then, the reactor was heated from 220 to 350 °C with a ramp rate of 10 °C/min and held for 2 h under regeneration gas. Regeneration gas contained 10% O₂, 5% CO₂, 10% H₂O balanced with N₂. **b**, Selected FT-IR spectra during regeneration of catalyst.

In Supplementary Fig. 6, we directly observed the gaseous species released from 100 ppm SO₂-aged VWTi catalyst during the regeneration step. Desorption of NH₃ was first observed, followed by desorption of SO₄²⁻ centered at 1115 cm⁻¹ (M. Hallquist et al., *Phys. Chem. Chem. Phys.* **5**, 3453-3463 (2003)). Desorption of SO₄²⁻ continued until 80 minutes during the regeneration at 350 °C. Such results directly prove that ABS present in the catalyst can decompose even at 350 °C under our regeneration condition, unlike inert condition.

Fig. R5. VWTi and VWTi+Y zeolite catalysts were aged under 500 ppm NO, 600 ppm NH₃, 10% O₂, 5% CO₂, 10% H₂O, 30 ppm SO₂ and balance N₂. Then, the catalysts were regenerated by thermal treatment at 270, 310, and 350 °C for 2 h, respectively. After regeneration at each temperature, the activity of catalyst was measured at 220 °C. Regeneration condition was 10% O₂, 5% CO₂, 10% H₂O and balance N₂.

Furthermore, we conducted additional experiments to regenerate VWTi and VWTi+Y zeolite catalysts at much lower temperature at 270 and 310 °C, respectively, as shown in Fig. R5. It was observed that the activity of catalyst gradually recovered as the regeneration temperature increased. This result is quite unexpected since ammonium bisulfate do not decompose sufficiently to restore the catalytic activity at low temperatures (270 and 310 °C). Hence, more detailed research about regeneration process is needed, but it can be concluded that in our reaction conditions, a temperature of 350 °C is required to completely decompose ABS on the catalyst.

Revised manuscript (page 6, line 122): We directly observed desorption of sulfate species released from SO₂-aged VWTi catalyst during the regeneration step (Supplementary Fig. 6).

Revised manuscript (page 8, line 166): Also, ammonium bisulfate species decompose much more easily under regeneration conditions (H₂O, O₂, CO₂/N₂) than under N₂ conditions, so the catalyst can be regenerated at 350 °C (Supplementary Fig. 6).

[Reviewer #3]

This is a very strong and novel paper, where it is shown that the addition of zeolite to a vanadia catalyst can reduce the sulfur poisoning. I recommend that the paper is considered for publication after updating the paper according to the comments below:

Question 1. On line 121 it is proposed that liquid ABS is formed. This statement should be further backed-up with experiments, or in more detail explain which experimental results that shows this.

Answer: Thank you for the valuable comment. The minimum temperature of NH_3 -SCR operation is determined by the formation of ammonium salts. Below the dew point, NH_3 and H_2SO_4 are known to condense into liquid ammonium bisulfate in the porous structure of catalysts, thereby degrading the catalytic performance. Although the formation of ammonium sulfate is thermodynamically more favorable than that of ammonium bisulfate, previous reports showed that ammonium bisulfate is dominantly formed due to kinetic limitations (J. Ando et al., *NOx Abatement for Stationary Sources in Japan*. EPA-600/7-79-205 EPA Contract No. 68-02-2161, (1979); J. R. Thøgersen, *Ammonium bisulphate inhibition of SCR catalysts*. Haldor Topsøe Inc.: Frederikssund, Denmark (2010)). In the case of VWTi+Z catalysts in our work, however, there is a possibility that ABS is not initially formed in the VWTi particle, but rather SO_3 reacts directly with NH_3 and H_2O in the pore of zeolite to form ABS. To clarify this, SO_2 deactivation behavior of physically mixed VWTi+Z catalyst was compared with that of loose-contacted 'VWTi+Z PM L' in Supplementary Fig. 10b.

Supplementary Fig. 10.

Comparison of physically mixed VWTi and Y zeolite (VWTi+Z) to loose contact sample (VWTi+Z PM L).

b, 22 h SO_2 aging profiles of VWTi, VWTi+Z, and VWTi+Z PM L under NH_3 -SCR reaction condition with 30 ppm SO_2 at 220 °C. NH_3 -SCR condition was under 500 ppm NO, 600 ppm

NH₃, 10% O₂, 5% CO₂, 10% H₂O and balance N₂. GHSV: 150,000 mL/h·g_{cat} based on VWTi weight.

Fig. R4. TEM image and line-EDS of VWTi+Z PM L catalyst after SO₂ aging.

The physically contacted area of VWTi and zeolite is very low in 'VWTi+Z PM L' catalyst, because VWTi and zeolite are individually sieved into particles and then mixed. It can be seen that 'VWTi+Z PM L' catalyst showed a similar deactivation rate to that of pure VWTi, which means that gaseous SO₃ cannot be captured directly by zeolite. Also in EDS analysis of SO₂-aged 'VWTi+Z PM L' catalyst, sulfur species are dominantly present on vanadia particle, not on zeolite (Fig. R4). The effect of alleviating sulfur deactivation is observed only in a well-mixed physical mixture, because the contact between VWTi and zeolite is necessary to trap liquid ammonium bisulfate formed on VWTi particle.

Revised manuscript (page 10, line 206): Also, SO₂ deactivation behavior of physically mixed VWTi+Z catalyst was compared with that of loose-contacted VWTi+Z PM L sample (Supplementary Fig. 10b). It can be seen that VWTi+Z PM L sample shows a similar deactivation rate to that of pure VWTi, demonstrating that gaseous SO₃ cannot be captured directly by zeolite. This result suggests that mixed H-Y zeolite could not work to alleviate sulfur deactivation without close physical contact to VWTi, likely because the H-Y zeolite traps sulfur via liquid phase ABS, not by gas phase SO₂ or SO₃.

Question 2. The sulfur is released with a main peak around 415C, see Fig 3 supp. However, the regeneration is performed at 350C. How can the regeneration be performed at a lower temperature than the sulfur release? Also the mixed vanadia/zeolite catalyst exhibit SO₂ desorption at 415 and 715C. There is a risk that the catalyst would be saturated with sulfur which would result in a degradation and that it is no longer possible to regenerate the catalyst at 350C. An experiment where the catalyst is saturated with sulfur, could for example be done by repeated cycles with much higher SO₂ concentration in order to examine that the regeneration still is functioning.

Answer: Thank you for the important comment. First, it should be noted that the regeneration of catalyst at 350 °C is only possible under oxidative conditions. We found that decomposition of ABS is largely different under inert gas and oxidative gas. According to our observations, sulfur species desorbed in the form of SO₂ under N₂, but they desorbed in the form of various aerosol including sulfate ion (SO₄²⁻) under O₂/N₂ flow. In the latter case, it is difficult to detect and quantify the sulfur species with FT-IR gas cell, and this is why we plotted thermal decomposition profiles of S-aged catalyst under inert flow (N₂) to clearly separate ammonium bisulfate and aluminum sulfate from the TPD data (Fig. 2f).

Fig. 2f. Thermal decomposition profiles of sulfur species on the SO₂-aged VWTi+Z and VWTi catalysts.

Under real regeneration conditions (H₂O, O₂, CO₂/N₂), however, ammonium bisulfate species decompose much more easily than under N₂ conditions, so the catalyst can be regenerated at 350 °C, which will be described in the following.

In Supplementary Fig. 6, we directly observed the gaseous species released from 100 ppm SO₂-aged VWTi catalyst during the regeneration step. Desorption of NH₃ was first observed, followed by desorption of SO₄²⁻ centered at 1115 cm⁻¹ (M. Hallquist et al., *Phys. Chem. Chem. Phys.* **5**, 3453-3463 (2003)). Desorption of SO₄²⁻ continued until 80 minutes in the regeneration at 350 °C. Such results directly prove that ABS present in the catalyst can decompose even at 350 °C under regeneration condition, unlike inert condition.

Supplementary Fig. 6.

Outlet gas profiles of VWTi catalyst during regeneration step.

a, Ammonia and sulfate profiles during regeneration step which were obtained by using FT-IR with a gas cell. Before regeneration, VWTi catalyst was aged under SCR reaction gas containing 100 ppm SO₂ at 220 °C for 22 h. Then, the reactor was heated from 220 to 350 °C with a ramp rate of 10 °C/min and held for 2 h under regeneration gas. Regeneration gas contained 10% O₂, 5% CO₂, 10% H₂O balanced with N₂. **b**, Selected FT-IR spectra during regeneration of catalyst.

The reviewer also wondered whether regeneration is still possible after the mixed zeolite is saturated with sulfur species. To confirm this, aging experiments were conducted by increasing the SO₂ concentration from 30 to 100 ppm for VWTi and VWTi+Sulfated Z catalysts as shown in Supplementary Fig. 9. About 60 wt.% of ABS was impregnated on Y zeolite (Si/Al₂=12) and calcined at 500 °C for 4h to prepare ‘Sulfated Z’. Based on elemental analysis, the amount of sulfur was 9.03 wt.%, which is 1.43 as S/Al ratio indicating that Al sites of the Y zeolite were nearly saturated by sulfur (Supplementary Fig. 8).

Supplementary Fig. 9.

Regeneration and reusability test of VWTi catalyst after SO₂ aging.

The catalyst was aged under 500 ppm NO, 600 ppm NH₃, 10% O₂, 5% CO₂, 10% H₂O, 30 ppm SO₂ and balance N₂. Regeneration condition was 10% O₂, 5% CO₂, 10% H₂O and balance N₂. VWTi catalyst was totally regenerated under the 350 °C thermal treatment.

It can be seen that ABS sorption function of ‘Sulfated Y zeolite’ still works even under 100 ppm SO₂ condition. Initial catalytic activity and lowered deactivation rate were also maintained after regenerating catalyst at 350 °C. This demonstrates that VWTi+Z catalyst can be used almost permanently through repeated regeneration.

Revised manuscript (page 6, line 122): We directly observed desorption of sulfate species released from SO₂-aged VWTi catalyst during the regeneration step (Supplementary Fig. 6).

Revised manuscript (page 8, line 166): Also, ammonium bisulfate species decompose much more easily under regeneration conditions (H₂O, O₂, CO₂/N₂) than under N₂ conditions, so the catalyst can be regenerated at 350 °C (Supplementary Fig. 6).

Revised manuscript (page 9, line 176): In addition, the aging experiment was repeated by increasing the SO₂ concentration from 30 to 100 ppm for VWTi+sulfur-saturated zeolite to confirm whether the regeneration is still possible (Supplementary Fig. 9). Initial catalytic activity and lowered deactivation rate were also maintained after regenerating catalyst at 350 °C, indicating that ABS sorption function of zeolite still works even after saturation with sulfur.

Question 3. The kinetic model is more simplified simulations for investigating a reaction mechanism. A kinetic model should include the effect of temperature. It is also assumed in the model that the rate is not depending on the SO₂ concentration, which is quite probable. If it is should be stated that a kinetic model is performed it must be done in much more detail. Alternatively, do not claim that it is a kinetic model but instead reaction mechanism investigations.

Answer: Thank you for the important advice. In Fig. 4, our aim was to find the factors that determine the ABS trapping ability of various zeolites. As the reviewer suggested, the expression of 'kinetic modelling' was changed to 'reaction mechanism investigation' for the entire manuscript.

Revised manuscript (page 12, line 253): With these two factors in mind, we hypothesize that the rate of ABS migration is a first-order function of the amount of framework Al (C_{AlO}), and we propose a mechanism of the migration by introducing several assumptions (the details of each of the assumptions are in the Methods section).

Revised manuscript (page 14, line 280): Reaction mechanism study demonstrates that the trapping ability of zeolites is dependent on the amount of framework Al and pore opening of structure that determine ABS migration rate.

Revised manuscript (page 18, line 368): Reaction mechanism investigation of ABS trapping was performed by using two determination factors; the amount of Al and the zeolite structure.

Question 4. Figure 2: Use real distance on x-axis instead of arbitrary units.

Answer: Thank you for the comment. We changed arbitrary units to real distance on x-axis in Fig. 2 in the revised manuscript.

Fig. 2. Location and species of sulfur after SO₂ aging. a, b, TEM image and line-EDS of the SO₂ aged VWTi. c, d, TEM image and line-EDS of the SO₂ aged VWTi+Z. e, Pore size distribution for various aging conditions. f, SO₂ partial pressure and temperature versus elapsed time for different aging conditions.

Question 5. Clarify briefly in the text in the results section regarding concentration during poisoning and regeneration to facilitate the reading.

Answer: Thank you for your comment. The concentration of used reactant was added for all the reaction experiments in the revised manuscript.

Revised manuscript (page 5, line 103): The laboratory reaction system simulated the off-gas emitted from a sintering plant by containing 500 ppm NO, 5% CO₂, 10% H₂O, 30 ppm SO₂ and 10% O₂ balanced with N₂, while 600 ppm NH₃ was introduced as reductant.

Revised manuscript (page 6, line 121): Regeneration gas contains 10% O₂, 5% CO₂, and 10% H₂O balanced with N₂.

Revised manuscript (page 28, line 597): Reaction gas contains 500 ppm NO, 500 ppm NH₃, 10% O₂, 5% CO₂, and 10% H₂O balanced with N₂;

REVIEWER COMMENTS

Reviewer #1 (Remarks to the Author):

After providing more detailed experimental results, the authors have addressed some issues. However, this study does not meet the level of this journal. I still have some more concerns on the innovation of this work and some other issues.

1. In recent years, much progress has been made in SO₂ resistance of zeolite mixed metal oxides catalysts for NH₃-SCR, e. g. Chemical Engineering Journal 316 (2017) 1059–1068; Journal of Alloys and Compounds 726 (2017) 906–912. Among these works, a lot of effort was paid in suppressing the adsorption and oxidization of SO₂ on catalysts which is the root problem of SO₂ poisoning. In this work, the authors claimed that physically mixing zeolites with VWTi to make NH₄HSO₄ (ABS) prefer to migrate from VWTi to zeolites. However, for industrial application, although ABS can get decomposed through reheating the catalyst, the high oxidization of SO₂ by VWTi still exist and NH₃ decomposed from ABS could continue to react with SO₃ to make ABS deposit in the pipeline at the back-end which can corrode equipments and increase the costing. Furthermore, the Al/Si ratio was thought to be in proportion to enhanced sulfur resistance. As reported in Chemical Engineering Journal 262 (2015) 1199–1207, the increased Al/Si ratio could strengthen the hydrophilicity of zeolites. Hence, could it be possible that the increased hydrophilicity is the key factor for the migration of ABS? If that is the case, the innovation of this work could not meet the level of the journal.

2. For the answer to the first question, compared with conventional vanadia/titania catalyst with 2 wt.% of loading amount, the content of vanadia of catalysts used in this work was increased to 5 wt.%, which had a better low-temperature activity. However, due to the severe biotoxicity of vanadia, much more efforts should be paid in developing catalysts with lower loading amount or even vanadia-free catalysts. Hence, the starting point and strategy in this work might mislead the further research direction in this field. Likewise, in the content of the answer to the first question, conventional catalyst (2VWTi) showed only 24% of NO_x conversion but the sulfur resistance was thought to be acceptable (Figure R1).

However, the steady activity during 22h SO₂ aging should be due to the low oxidizability of 2VWTi for SO₃ oxidized from SO₂ at 220 °C which should not be considered as a poor catalytic capacity instead of a great sulfur resistance. Please modify the description of sulfur resistance of the conventional catalyst.

3. For the answer to the second question, you performed the SO₂ aging under the condition with much higher SO₂ concentration of 100 ppm (Supplementary Figure 4). However, the normalized activity of two catalysts before the addition of SO₂ was different. Was this caused by instrumental error or the poor repeatability of the mechanical mixing of VWTi+Y?

4. For the answer to Question 3 and 4, Arrhenius plot of VWTi with 2 wt.% of ABS was displayed in Supplementary Fig. 11. Please explain why VWTi with lower amount of ABS was chosen instead of samples tested in the body of paper.

5. In Supplementary Fig. 11a, the unit in the title of top of diagram was K. However, according to the discussion, it should be °C. Please modify the diagram and recheck all figures in this paper.

6. In Supplementary Fig. 11b, values of the activation energy of both samples got decreased to 61.1 and 58.3 above 160 °C, respectively. However, based on the discussion in this part, the activation energy should be unchanged. Please supplement the explanation for the decrease. Similarly, in Supplementary

Fig. 11b, the pre-exponential factors of both two catalysts got decreased above the melting point of ABS. However, as the discussion in this paper, ABS prefers to migrate to zeolites to prevent VWTi which should cause the increase of the pre-exponential factor for VWTi+Z. Please give your explanation for this phenomenon.

7. In the answer to Question 3 and 4, above the melting point of ABS, VWTi+Z and VWTi+Z PM L had similar activation energy but the pre-exponential factor of the PM L sample was thought to got decreased from 1.8×10^6 to 6.7×10^5 . However, the pre-exponential factor of two different samples should not be compared with each other. Hence, this part should be modified that the change of the pre-exponential factor of VWTi+Z from 2.1×10^7 to 1.8×10^6 was slighter than that of VWTi+Z PM L from 2.0×10^7 to 6.7×10^5 , which showed that the ABS prefers giving rise to physical deactivation in the PM L catalyst.

8. In the answer to Question 5, two resonance peaks at 55 ppm and 0 ppm in Supplementary Fig 4a are ascribed to framework Al with tetrahedral coordination and extra-framework Al with octahedral coordination. Please supplement relevant references to support the assignments. Furthermore, NH₃-TPD was applied to investigated the acid properties that two desorption peaks are indiscreetly assigned to NH₃ on Lewis acid sites and Brønsted acid sites, respectively, which was ill-considered. Please think over the discussion of this part and supplement the NH₃ signal of NH₃-TPD combined with mass spectrometry to indentify the NH₃ desorption behavior more precisely.

Reviewer #2 (Remarks to the Author):

The manuscript has been improved. My questions become clear.

Reviewer #3 (Remarks to the Author):

The authors have well addressed my comments. I recommend to accept the paper for publication.

Response to the reviewers' comments

Sincerest thanks for your response and referees' valuable advice and comments on our manuscript. To fully address the reviewers' comments, the additional experiments were conducted and manuscript was carefully revised. Details are shown below, including a point by point response to the reviewers' comments marked in blue and newly added sentences in the revised manuscript marked in red, and we hope that the revisions will comply with the referees' remarks. We hope that a revised version of the manuscript will be processed in due course by *Nature Communications*.

[Reviewer #1]

After providing more detailed experimental results, the authors have addressed some issues. However, this study does not meet the level of this journal. I still have some more concerns on the innovation of this work and some other issues.

Question 1. In recent years, much progress has been made in SO₂ resistance of zeolite mixed metal oxides catalysts for NH₃-SCR, e. g. Chemical Engineering Journal 316 (2017) 1059–1068; Journal of Alloys and Compounds 726 (2017) 906–912. Among these works, a lot of effort was paid in suppressing the adsorption and oxidization of SO₂ on catalysts which is the root problem of SO₂ poisoning. In this work, the authors claimed that physically mixing zeolites with VWTi to make NH₄HSO₄ (ABS) prefer to migrate from VWTi to zeolites. However, for industrial application, although ABS can get decomposed through reheating the catalyst, the high oxidization of SO₂ by VWTi still exist and NH₃ decomposed from ABS could continue to react with SO₃ to make ABS deposit in the pipeline at the back-end which can corrode equipments and increase the costing. Furthermore, the Al/Si ratio was thought to be in proportion to enhanced sulfur resistance. As reported in Chemical Engineering Journal 262 (2015) 1199–1207, the increased Al/Si ratio could strengthen the hydrophilicity of zeolites. Hence, could it be possible that the increased hydrophilicity is the key factor for the migration of ABS? If that is the case, the innovation of this work could not meet the level of the journal.

Answer: We thank the reviewer for valuable comments. As the reviewer pointed out, the fundamental problem of SO₂ poisoning consists of chemical poisoning, where the catalyst lose its activity as the active sites are chemically deactivated by SO₂, and physical poisoning, where the access of reactants are inhibited as the layer of ammonium bisulfate (ABS) is formed on catalyst surface. We addressed both poisoning phenomena in this work, and we were able to explain the difference and originality of our results compared to other papers as follows.

The main reason for using transition metals such as Mn, Fe, and Cu instead of V is to lower the operating temperature of the SCR reaction to the low-temperature region. Although numerous alternative catalysts have been reported so far, chemical poisoning with SO₂ has

not been solved yet. Only vanadium-based catalysts are widely used in industry because they are relatively free from chemical poisoning with SO₂ (L. Xu et al., *Environ. Sci. Technol.* **52**, 7064-7071 (2018)). In the paper mentioned by the reviewer (C. Yu et al., *Chem. Eng. J.* **316**, 1059-1068 (2017)), it can be seen that the catalytic activity sharply decreases in 1 hour when SO₂ is injected at 200 °C, which is typical phenomena of chemical poisoning by SO₂. (Note that V₂O₅-WO₃/TiO₂ catalyst does not exhibit a sharp decrease in activity when exposed to SO₂ as described in our work, and that is why V-based catalyst can be widely utilized.) The improvement of SO₂ resistance in the papers mentioned by the reviewer is that they can alleviate the degree of chemical poisoning, not the ABS formation problem. For example, the addition of Pr on MnO_x/SAPO-34 catalyst can alleviate the sharp decrease in activity when exposed to SO₂, but the formation of Pr sulfate also degrade the redox ability of catalyst that can lead to irreversible deactivation. In order to decompose the formed metal sulfate, a high temperature of 800-900 °C is required, as described in their paper, and chemical poisoning cannot be recovered reversibly during SCR operation. Most of the low-temperature catalysts containing Mn, Fe, and Ce including Cu-zeolite are not free from chemical poisoning with SO₂ and are not easy to be applied to the industry (S. Dahlin et al., *Catal. Today* **320**, 72-83 (2019); L. Xu et al., *Environ. Sci. Technol.* **52**, 7064-7071 (2018)). Therefore, we increased the low-temperature activity of V₂O₅-WO₃/TiO₂ catalyst by increasing the V content for application to the industry, which is free from chemical deactivation by SO₂. As the reviewer mentioned, even with our developed catalyst, it is hard to completely prevent the corrosion of pipeline. However, considering the performance and lifetime of the SCR catalyst, V₂O₅-WO₃/TiO₂ is irreplaceable catalyst that requires further development.

An important problem in the low-temperature operation of V-based catalysts is physical deactivation originating from the formation of ABS. To the best of our knowledge, our work is the first to propose a method of physically mixing H-form zeolite to solve this problem. The reviewer claimed that zeolite was already mixed with metal oxide catalysts to improve SO₂ resistance in other studies, but as a result of our careful reading of the papers, we can see that these papers did not specifically deal with the ABS deactivation problem. For example, C. Yu et al. proposed MnO_x catalyst that dispersed on the SAPO-34 zeolite, a system completely different from the physical mixture catalyst (C. Yu et al., *Chem. Eng. J.* **316**, 1059-1068 (2017)). In addition, the formation of ABS is not clear because H₂O was not added during the SO₂ aging procedure. ABS is not formed without water, and therefore the effect of zeolites on ABS deactivation cannot be confirmed in their studies. They just used SAPO-34 as a support to disperse Mn oxides well. Meanwhile, B. Chi et al. proposed a core shell of CeO₂@Fe-ZSM-5 to prevent direct contact between SO₂ and CeO₂ catalyst (B. Chi et al., *J. Alloy. Compd.* **726**, 906-912 (2017)). However, the SCR activity decreased when only SO₂ is added over CeO₂@Fe-ZSM-5 catalyst, indicating that the presence of zeolite shell did not completely prevent the chemical deactivation by SO₂. In particular, SO₂ aging was performed at 300 °C, a relatively high temperature at which ABS is not well formed. In order to simulate ABS deactivation, SO₂ and H₂O should be flowed simultaneously at a low temperature below 250 °C, lower than the dew point of ABS. In other words, they did not use H-form zeolite and did not simulate ABS deactivation, thus their purpose was completely

different from our study.

Moreover, all of the above studies only confirmed that the developed catalyst was less deactivated and did not confirm whether the deactivated catalyst could be regenerated. We proved in our work that physically mixed catalyst can be regenerated at 350 °C because it traps SO₃ in the form of ammonium sulfate, not metal sulfate species. The reviewer questioned whether there was a relationship between the ABS trapping ability and the hydrophilicity of the mixed zeolites. According to our experiments and DFT calculations, Brønsted acid sites in zeolite can significantly lower the adsorption energy of ABS coordinated with water or NH₃. We proposed that this is the reason why the higher the Al/Si ratio of mixed zeolites, the higher the ABS trapping ability. It has been well known that the hydrophilicity of zeolite increases with the Al/Si ratio, but we do not yet know the direct relationship between ABS trapping ability and hydrophilicity of zeolite, and further studies will be needed. T. Du et al. introduced silicate shell into Fe-ZSM-5 to control hydrophobicity and increase the activity of NH₃-SCR, while the experiments related to ABS deactivation were not conducted (T. Du et al., *Chem. Eng. J.* **262**, 1199-1207 (2015)).

In summary, the innovation of our work is to propose a simple method of physically mixing zeolite with vanadium catalyst to prevent not only the physical ABS deactivation, but also the chemical deactivation, which has never been reported. We believe that our work is novel in discovering the phenomenon of trapping ammonium bisulfate, a deactivating substance, by physically mixed zeolite particles.

Question 2. For the answer to the first question, compared with conventional vanadia/titania catalyst with 2 wt.% of loading amount, the content of vanadia of catalysts used in this work was increased to 5 wt.%, which had a better low-temperature activity. However, due to the severe biotoxicity of vanadia, much more efforts should be paid in developing catalysts with lower loading amount or even vanadia-free catalysts. Hence, the starting point and strategy in this work might mislead the further research direction in this field. Likewise, in the content of the answer to the first question, conventional catalyst (2VWTi) showed only 24% of NO_x conversion but the sulfur resistance was thought to be acceptable (Figure R1). However, the steady activity during 22h SO₂ aging should be due to the low oxidizability of 2VWTi for SO₃ oxidized from SO₂ at 220 °C which should not be considered as a poor catalytic capacity instead of a great sulfur resistance. Please modify the description of sulfur resistance of the conventional catalyst.

Answer: Thank you for the important comments. We fully agree that future research should aim at lowering the content of toxic vanadium oxides as the reviewer pointed out. However, there are still many fields where vanadium catalysts should be used as environmental regulations are tightening much faster than the pace of technological progress on the development of vanadium-free SCR catalysts. For example, Cu-zeolite catalysts cannot be applied yet to SCR equipment in steel-making process, because the exhaust gas contains 30 ppm of SO₂ that can rapidly degrade the catalytic activity. That is why we considered how to

use V-based catalyst for the low-temperature SCR for rapid commercialization. The results of this work are expected to be useful in many applications that burn fuels containing sulfur.

In addition, we already mentioned in our work that the degree of ABS deactivation is proportional to the amount of SO_3 produced by SO_2 oxidation. Although the SO_2 oxidation ability may vary depending on the type of catalysts, most SCR catalysts are thought to have SO_2 oxidation ability to some extent. This is because low-temperature SCR reaction requires a catalyst with good redox property. That is, the physical deactivation by ABS formation is a phenomenon that can occur in the same manner not only in vanadium catalyst but also in other catalysts (K. Guo et al., *Chem. Eng. J.* **389**, 124271 (2020)). The method of trapping ABS with a physically mixed zeolite that we have proposed can be applied to other metal oxide catalysts. Therefore, since this strategy is not limited to vanadium catalyst, we expect that it can be utilized complementarily with other research on developing environmental SCR catalyst in the future.

Revised manuscript (page 13, line 280): Because this strategy is not limited to vanadium catalyst, we expect that it can be utilized complementarily with other research on developing environmental SCR catalyst in the future.

Question 3. For the answer to the second question, you performed the SO_2 aging under the condition with much higher SO_2 concentration of 100 ppm (Supplementary Fig. 4). However, the normalized activity of two catalysts before the addition of SO_2 was different. Was this caused by instrumental error or the poor repeatability of the mechanical mixing of VWTi+Y?

Supplementary Fig. 2.

Reaction test of VWTi and VWTi+Z.

a, Reactivity of VWTi and VWTi+Z at 100-450 °C under NH_3 -SCR condition (500 ppm NO, 600 ppm NH_3 , 10% O_2 , 5% CO_2 , 10% H_2O and balance N_2 , GHSV=150,000 mL/h g catalyst of VWTi).

Answer: Thank you for the valuable question. We are sorry for the typo error that the y-axis

in Supplementary Fig. 4 is NO_x conversion, not the normalized activity (It was revised.). We think that the slightly different catalytic activity measured at 220 °C is due to instrumental error. The VWTi catalyst used in our work was observed to show a rapid increase in activity in the temperature range between 150-250 °C (Supplementary Fig. 2a). Because the temperature of SO₂ aging (220 °C) is between these temperature ranges, it is expected that the measured activity can be affected even with a very small temperature oscillation. When we conducted several reaction experiments with the VWTi+Z catalyst, the catalyst always showed an initial NO_x conversion in the range of 62-66% at 220 °C consistently.

Question 4. For the answer to Question 3 and 4, Arrhenius plot of VWTi with 2 wt.% of ABS was displayed in Supplementary Fig. 11. Please explain why VWTi with lower amount of ABS was chosen instead of samples tested in the body of paper.

Fig. R1. a, Arrhenius plots of 10 wt.% ABS/VWTi+Z (black dot) and 10 wt.% ABS/VWTi+Z PM L (red dot) during transient NH₃-SCR reaction (temperature from 100 °C to 220 °C with ramp rate 1 °C/min, other reaction conditions were the same as steady-state NH₃-SCR experiment. b, Same experiments were conducted by lowering the amount of ABS to 2 wt.%.

Answer: We thank the reviewer for the insightful comments. In the discussion about the absorptive protection mechanism in Fig. 3a in the manuscript (replotted in Fig.R1a), the change of the slope in the Arrhenius plots of the ABS/VWTi+Z PM L material is attributed to the physical deactivation by the phase transformation of ABS from solid to liquid over the VWTi catalyst. Because this is physical deactivation, the apparent activation energy should not be changed after ABS deactivation. However, the apparent activation energy was changed during ramping. We have discussed the results and assumed that the amount of ABS was too much (10 wt.%) to observe the expected result, which means that the migration of ABS might be still on-going even at 160~215 °C. Then, we conducted same experiments with 2 wt.% of ABS impregnated VWTi that is much lower amount, and Arrhenius plot is displayed in Supplementary Fig. 11 (replotted in Fig. R1b). Above the melting point of ABS (160 °C), VWTi+Z and VWTi+Z PM L show same slopes of Arrhenius plots (61.1 kJ/mol and 58.3

kJ/mol, respectively), instead the pre-exponential factor decreased in the PM L sample ($1.8 \times 10^6 \rightarrow 6.7 \times 10^5$). This result clearly illustrates that the ABS gives rise to physical deactivation in the PM L catalyst. The results over 10 wt.% ABS loaded samples were used in the revised manuscript for clarity, because the difference between the two samples, ‘VWTi+Z’ and ‘VWTi+Z PM L’, is more pronounced above melting point of ABS. Such observations clearly reveal the importance of physical contact between VWTi and zeolite particles in ABS trapping.

Question 5. In Supplementary Fig. 11a, the unit in the title of top of diagram was K. However, according to the discussion, it should be °C. Please modify the diagram and recheck all figures in this paper.

Supplementary Fig. 11.

a, Reactivity of VWTi and VWTi+Z at 100-450 °C under NH₃-SCR condition (500 ppm NO, 600 ppm NH₃, 10% O₂, 5% CO₂, 10% H₂O and balance N₂, GHSV=150,000 mL/h g catalyst of VWTi). **b**, Activation energy (E_a) and pre-exponential factor obtained from the Arrhenius plots of 2wt ABS/VWTi+Z and 2wt ABS/VWTi+Z PM L during the transient SCR reaction.

Answer: Thank you for your comments. We corrected the title of X axis of Fig. S11a to °C and rechecked all figures in the manuscript.

Question 6 and 7. In Supplementary Fig. 11b, values of the activation energy of both samples got decreased to 61.1 and 58.3 above 160 °C, respectively. However, based on the discussion in this part, the activation energy should be unchanged. Please supplement the

explanation for the decrease. Similarly, in Supplementary Fig. 11b, the pre-exponential factors of both two catalysts got decreased above the melting point of ABS. However, as the discussion in this paper, ABS prefers to migrate to zeolites to prevent VWTi which should cause the increase of the pre-exponential factor for VWTi+Z. Please give your explanation for this phenomenon.

In the answer to Question 3 and 4, above the melting point of ABS, VWTi+Z and VWTi+Z PM L had similar activation energy but the pre-exponential factor of the PM L sample was thought to get decreased from 1.8×10^6 to 6.7×10^5 . However, the pre-exponential factor of two different samples should not be compared with each other. Hence, this part should be modified that the change of the pre-exponential factor of VWTi+Z from 2.1×10^7 to 1.8×10^6 was slighter than that of VWTi+Z PM L from 2.0×10^7 to 6.7×10^5 , which showed that the ABS prefers giving rise to physical deactivation in the PM L catalyst.

Answer: Thank you for the very important comments. The deposited ABS was known to deactivate the catalyst as the form of sticky liquid blocking the pore, which can be inferred that solid ABS could not act as a deactivating materials below the melting point. Therefore, the pre-exponential factor should decrease after 160 °C in the transient reaction of 2wt ABS/VWTi+Y PM L like our data in supplementary Fig. 11b since the ABS on the VWTi started to deactivate the catalyst as it transformed to liquid phase at 160 °C. And then, the profiles of 2wt ABS/VWTi+Y should not change by ABS because there was no deactivating material on VWTi as the ABS migrated to the Y zeolite.

Fig. R2. The Arrhenius plot of transient SCR reaction on VWTi catalyst.

However, as the reviewer pointed out, there were difference between the values obtained below 160 °C and above 160 °C even in the 2wt ABS/VWTi+Y sample. It might be because the obtained Arrhenius plots of 2wt ABS/VWTi+Z did not show perfect linearity, and it can be attributed to following reason. The reaction rate was not measured under perfect

steady state condition because it was transient experiment although we ramped up the temperature very slowly. As the reaction rate was getting higher in increasing temperature, the measured reaction rate deviates from the actual steady state rate, which made the activation energy underestimated. This phenomenon is also observed at the VWTi catalyst (Fig. R2), which ensures that it did not result from effect of ABS impregnation or physical mixing of zeolite. The pre-exponential factors were also measured lower with the same reason because they were obtained from the extrapolated y-intercept of the graph, which must have been underestimated by the decrease of the slope. Therefore, it is difficult to investigate the change in E_a and A depending on the temperature, so instead, we compared the values from 2wt ABS/VWTi+Y and 2wt ABS/VWTi+Y PM L within same temperature range. We postulate that it is reasonable because the SCR reaction was occurred at the same active material (VWTi) in both catalysts.

Revised manuscript (Supplementary Fig. 11): Below the melting point of ABS (160 °C), the 2wt ABS/VWTi+Z and 2wt ABS/VWTi+Z PM L shows same activation energy ($E_a = 70.4$ kJ/mol) and pre-exponential factor values ($A = 2.1 \times 10^7$ and 2.0×10^7) because VWTi catalyst cannot be deactivated when impregnated ABS is present as solid. Above 160 °C, the both catalysts still show almost same E_a value of 61.1 and 58.3 kJ/mol. However, the pre-exponential factor (A) decreased from 1.8×10^6 to 6.7×10^5 in the 2wt ABS/VWTi+Y PM L, which clearly indicates that the impregnated ABS physically deactivated the VWTi catalyst.

According to Arrhenius equation, the E_a value obtained below 160 °C and above 160 °C should have been the same, but there is slight difference (70.4 kJ/mol and 61.1 kJ/mol, respectively). This did not result from effect of impregnated ABS or physical mixed Y zeolite, since same phenomenon was observed in Arrhenius plot of VWTi (data was not shown). We suppose that such discrepancy results from the gap between steady state reaction and transient reaction. As these experiments were conducted under a transient protocol, the Arrhenius plots obtained would not show perfect linearity.

Question 8. In the answer to Question 5, two resonance peaks at 55 ppm and 0 ppm in Supplementary Fig 4a are ascribed to framework Al with tetrahedral coordination and extra-framework Al with octahedral coordination. Please supplement relevant references to support the assignments. Furthermore, NH_3 -TPD was applied to investigated the acid properties that two desorption peaks are indiscreetly assigned to NH_3 on Lewis acid sites and Brønsted acid sites, respectively, which was ill-considered. Please think over the discussion of this part and supplement the NH_3 signal of NH_3 -TPD combined with mass spectrometry to identify the NH_3 desorption behavior more precisely.

Fig. R3. NH₃ desorption profiles of Y zeolite and S saturated Y zeolite during NH₃-TPD detected with a mass spectroscopy.

Supplementary Fig. 14. Characteristics of Y zeolite and Sulfated Y zeolite.

a, ²⁷Al-NMR spectra of Y zeolite and Sulfated Y zeolite. b, NH₃-TPD profiles of Y zeolite and Sulfated Y zeolite. NH₃ adsorbed at 50 °C for NH₃ saturation and purged under He. Then, the samples were ramped to 500 °C with the rate of 5 °C/min.

Answer: Thank you for your comments. We analyzed the ^{27}Al -NMR spectra referring to several literatures. As the reviewer pointed out, we add following references in our revised manuscript (S. J. Schmieg et al., *Catal. Today* **184**, 252-261 (2012); D. Coster et al., *J. Phys. Chem.* **98**, 6201-6211 (1994)).

Fig. R3 demonstrates the mass spectroscopy profiles used to analyze the desorbed NH_3 during the NH_3 -TPD protocol. The mass profiles ($m/z=16$) is almost identical to TCD signal in Supplementary Fig. 14. Two clear desorption peaks at low and high temperature are detected in profiles that we assigned them to NH_3 adsorbed on Lewis acid and Brønsted acid sites, respectively. This is a general assignment, which has been prevalently accepted among a lot of researchers (J. R. D. Iorio et al., *Topics in Catal.* **58**, 424-434 (2015); Y. Gao, et al., *RSC adv.* **6**, 83581-83588 (2016)). Especially, although there is some controversy on the origin of the low temperature NH_3 desorption peak, it is undoubtable to postulate that the high temperature peak is assigned to the strong Brønsted acid sites originated from the framework Al. Based on the such assignment, it can be concluded that the Brønsted acid sites of S saturated Y zeolite hardly change compared to the pristine Y zeolite.

Revised manuscript (Supplementary Fig. 14): The resonance peaks at 55 ppm and 0 ppm are assigned to framework Al and extra-framework Al, respectively (reference 8, 9 in supplementary information). The peak at 55 ppm decreased dramatically in sulfur saturated Y zeolite, which indicates the strong interaction between sulfate and framework Al. To investigate acidic properties of the Y zeolite and sulfur saturated Y zeolite, the NH_3 -TPD experiment was performed. It is well known that the Y zeolite shows two NH_3 desorption peaks at low and high temperature (120 and 300 °C, respectively) assigned to weakly bound NH_3 on Lewis acid sites and NH_3 on Brønsted acid sites, respectively. Although the sulfate was bound to framework Al, there was little change in the number of Brønsted acid sites in sulfur saturated Y zeolite compared to the pristine Y zeolite, probably because the sulfated sites can also act as the Brønsted acid sites. As a result, VWTi+Y catalyst demonstrates the almost same performance during a series of aging and regeneration cycle.

REVIEWER COMMENTS

Reviewer #1 (Remarks to the Author):

In the latest response, the authors have provided detailed explanation and supplemented some more experimental data to address my questions. However, there are still some serious issues on the discussion of key data in this paper. Furthermore, the innovation and significance of this work cannot meet the level of this journal. All in all, I don't think this work can be published.

1. SO₂-poisoning of SCR catalysts can be generally divided into three steps: (1) the adsorption of SO₂ at the initial stage of catalytic process, (2) the oxidation of SO₂ to SO₃, and (3) the physical deactivation by the deposition of (NH₄)₂SO₄/NH₄HSO₄ at the end of the whole process (L. Han et al., Chem. Rev. 119, 10916–10976 (2019)). Although there were some works focusing on the deposition of ABS in recent years and some interesting strategies were proposed (Y. Chen et al., Environ. Sci. Technol. 52, 11796–11802 (2018); K. Guo et al., ACS Appl. Mater. Interfaces 11, 4900–4907 (2019)), the root cause of this physical poisoning ought to be the high adsorption and oxidation SO₂ over metal oxides catalysts and was still unsolved in these works. In this paper, the migration mechanism of ammonium bisulfate by mixing zeolites and VWTi catalyst was illuminated, the same defect as mentioned before still existed that the adsorption of SO₂ was not discussed and the corrosion of pipeline by formed ABS was inevitable. It means that this work would not bring a significant guidance of solving SO₂-poisoning of SCR catalysts.

2. As mentioned in the answer to question 2, V based catalysts were thought to be irreplaceable in burn fuels containing sulfur because the development of vanadium-free catalysts could not meet the requirement of the latest environmental regulations. I totally agree with this opinion and understand this situation, but it cannot be a convincing reason to adopting vanadia/titania catalysts with a higher loading amount (5 wt. %) of V₂O₅ than the commercial catalyst. The higher amount of vanadium loading can also accelerate the oxidation of SO₂ to SO₃ (H. Kamata et al., Catalysis Letters 73, 79–83 (2001)) and then SO₃ can aggravate the corrosion of equipments by the formation of sulfates. Hence, although V based catalysts have a great advantage in burn fuels containing sulfur, the low-temperature activity of vanadia/titania catalysts should be improved by some other aspects such as modification of dispersion state of active sites, strengthen the interaction between metal oxides and support instead of simply increasing the amount of V₂O₅ and the research direction of SO₂-resistance of catalysts should also focus on the suppression of the SO₂ adsorption and oxidation.

3. In Supplementary Fig. 2, we can see that the NO_x conversion of VWTi+Z exceeded 100 % within the range from 250 °C to 300 °C, which is unreasonable. Please recheck the raw data and modify this figure.

4. For the answer to question 4, the phenomenon that the apparent activation energy of 10 wt.% ABS/VWTi+Z PM L got changed during ramping was thought be caused by the excess loading amount of ABS and the on-going migration of ABS. However, there was no relevant reference to support your assumption and the migration of excess ABS was still the physical behavior of deposit on the surface of catalysts, which should not change the apparent activation energy. Please provide some more reference and give a logical explanation.

5. There were some Arrhenius plots of catalysts obtained in this paper to describe the state of surface species. To avoid the mass transfer limitations, the Arrhenius plots should be measured at low

conversions (< 20%) (A. Wang et al., Nat Commun 10, 1137 (2019); J. Song et al., ACS Catal. 7, 12, 8214–8227 (2017)). However, in this work, the NO_x conversion of 10 wt.% ABS/VWTi+Z catalyst achieved 20 % at 150 °C while the temperature range of apparent energy tests was from 135 °C to 215 °C which could affect the reliability of some conclusions. What's more, the NO_x conversion data of catalysts with 2 wt.% ABS was not provided in this paper, which could be more easily to exceed 20 % of NO_x conversion.

6. There had the same issue on all figures of Arrhenius plots such as Fig. R1, Fig. R2 and Supplementary Fig. 11 that the scale of the bottom cannot be in line with that of the top. For example, for $1000/K=2.1$, the corresponding centigrade temperature should be 203 °C which was on the left of 200 °C instead of on the right of 200 °C on the top of figure. Please recheck all the pictures carefully and modify the description of these figure in this paper.

7. For the answer to question 6 and 7 and the revised manuscript, the name of catalysts were 2wt ABS/VWTi+Y and 2wt ABS/VWTi+Y PM L. However, the catalysts in original text were named as 2wt ABS/VWTi+Z and 2wt ABS/VWTi+Z PM L, which should be rechecked and modified.

8. For the answer to question 6 and 7, the decrease of the activation energy of both samples above 160 °C was explained by the gap between steady state reaction and transient reaction. If this explanation is correct, the discussion around the pre-exponential factor values of catalysts should also be inaccurate. Furthermore, the unperfect linearity Arrhenius plots of VWTi could be caused by the mass transfer limitations at a high conversion which has been mentioned above. Please learn more references and literatures around apparent energy test. These mistakes made in the discussion of key data were extremely serious. Hence, I suggest this paper should be rejected.

Response to the reviewer's comments

Sincerest thanks for your response and referee's valuable advice and comments on our manuscript. To fully address the reviewer's comments, the additional experiments were conducted and manuscript was carefully revised. Details are shown below, including a point by point response to the reviewer's comments marked in blue and newly added sentences in the revised manuscript marked in red, and we hope that the revisions will comply with the referees' remarks. We hope that a revised version of the manuscript will be processed in due course by *Nature Communications*.

[Reviewer #1]

In the latest response, the authors have provided detailed explanation and supplemented some more experimental data to address my questions. However, there are still some serious issues on the discussion of key data in this paper. Furthermore, the innovation and significance of this work cannot meet the level of this journal. All in all, I don't think this work can be published.

Question 1. SO₂-poisoning of SCR catalysts can be generally divided into three steps: (1) the adsorption of SO₂ at the initial stage of catalytic process, (2) the oxidation of SO₂ to SO₃, and (3) the physical deactivation by the deposition of (NH₄)₂SO₄/NH₄HSO₄ at the end of the whole process (L. Han et al., *Chem. Rev.* 119, 10916–10976 (2019)). Although there were some works focusing on the deposition of ABS in recent years and some interesting strategies were proposed (Y. Chen et al., *Environ. Sci. Technol.* 52, 11796–11802 (2018); K. Guo et al., *ACS Appl. Mater. Interfaces* 11, 4900–4907 (2019)), the root cause of this physical poisoning ought to be the high adsorption and oxidation SO₂ over metal oxides catalysts and was still unsolved in these works. In this paper, the migration mechanism of ammonium bisulfate by mixing zeolites and VWTi catalyst was illuminated, the same defect as mentioned before still existed that the adsorption of SO₂ was not discussed and the corrosion of pipeline by formed ABS was inevitable. It means that this work would not bring a significant guidance of solving SO₂-poisoning of SCR catalysts.

Answer: Thank you for the comment. Given the strengthened environmental regulations, preventing ABS deactivation while maintaining high activity at low temperatures is a very urgent issue in the field of SCR technology. However, it is hard to solve these two problems at the same time according to the studies reported so far (C. Liu et al., *Catalysts* 10, 1034 (2020)). Although the catalyst must have sufficient redox ability to have SCR activity at low temperatures, it may also accelerate the SO₂ oxidation reaction. That is why we decided to introduce ABS trapping site. To the best of our knowledge, our result is the first to meet the low-temperature SCR activity as well as to solve ABS deactivation, so that it is expected to attract a lot of interest from researchers in this field. In addition, we simply increased V loading to meet low-temperature activity without using any complex synthetic process, which

allows rapid commercialization. We agree that this approach didn't solve the root problem of SO₂ oxidation, but it is very helpful in solving the urgent environmental issues faced.

In the paper mentioned by reviewer (L. Han et al., *Chem. Rev.* 119, 10916–10976 (2019)), various methods have been tried to prevent ABS deactivation in addition to the method of preventing SO₂ oxidation, which include (1) alleviating the adsorption of SO₂ on catalyst, (2) improving the accessibility of reaction gas to catalyst surface in the presence of sulfate, (3) adding sacrificial sites to protect SCR active sites, (4) facilitating the decomposition of sulfate species, and (5) developing SO₂-tolerant material with high SCR activity. Our strategy utilizing zeolite as ABS trapping material has the advantage of facilitating the decomposition of sulfate species while being included in the category of solutions using sacrificial sites. In addition, the paper (L. Han et al., *Chem. Rev.* 119, 10916–10976 (2019)) indicated that there are still limitations in the study of mitigating SO₂ oxidation. For example, although the SO₂ oxidation ability of the catalyst can be suppressed to some extent, it is difficult to completely prevent the formation of SO₃ with current technology. This is because the catalysts active in the SCR reaction inevitably have redox ability, and in particular, NO₂ contained in the exhaust gas has a strong oxidizing ability and can promote SO₂ oxidation. That is, as long as a small amount of SO₃ is generated, the strategy of trapping SO₃ in the form of ABS using zeolite is always useful. Potential corrosion of the downstream pipeline can also be prevented by trapping the generated SO₃ in zeolite. The only problem in this system is that the stored SO₃ can be released again during the regeneration process. However, it is possible to explore other technical solutions to the pipeline corrosion issue, such as injecting an alkaline solvent at the rear end of catalyst bed during the regeneration process (D. Xie et al., *J. Chem. Technol. Biotechnol.* 94, 2382-2388 (2019)). It should be emphasized that the top priority of our research is to prevent the rapid deactivation of SCR catalysts by SO₃ while operating at low-temperatures.

Recent studies on SO₂-resistant SCR catalysts have focused on the usage of CeO₂ or Fe₂O₃ that retaining their activity even after sulfation. The sulfated CeO₂ facilitated the adsorption of ammonia, which promoted the SCR activity via the E-R mechanism (S. Yang et al., *Appl. Catal. B* 136, 19-28 (2013); L. Ma et al., *Appl. Catal. B* 232, 246-259 (2018)). The sulfated Fe₂O₃ catalysts also showed enhanced activity after sulfation above 275 °C (C. Tang et al., *Catal. Today* 307, 2-11 (2018)). However, it is stated that improving their low-temperature activity is quite intractable (L. Han et al., *Chem. Rev.* 119, 10916–10976 (2019)). Also, sulfation of CeO₂ is irreversible at low temperatures, limiting its regeneration in actual operation. Considering that the research using other metal oxides has reached its limit due to such a clear drawback, our trapping strategy can be a new breakthrough in developing SO₂-resistant catalyst.

Question 2. As mentioned in the answer to question 2, V based catalysts were thought to be irreplaceable in burn fuels containing sulfur because the development of vanadium-free catalysts could not meet the requirement of the latest environmental regulations. I totally agree with this opinion and understand this situation, but it cannot be a convincing reason to

adopting vanadia/titania catalysts with a higher loading amount (5 wt. %) of V_2O_5 than the commercial catalyst. The higher amount of vanadium loading can also accelerate the oxidation of SO_2 to SO_3 (H. Kamata et al., *Catalysis Letters* 73, 79–83 (2001)) and then SO_3 can aggravate the corrosion of equipments by the formation of sulfates. Hence, although V based catalysts have a great advantage in burn fuels containing sulfur, the low-temperature activity of vanadia/titania catalysts should be improved by some other aspects such as modification of dispersion state of active sites, strengthen the interaction between metal oxides and support instead of simply increasing the amount of V_2O_5 and the research direction of SO_2 -resistance of catalysts should also focus on the suppression of the SO_2 adsorption and oxidation.

Answer: Thank you for the important comment. We fully agree that several other methods must be devised to increase the low-temperature activity of V-based catalyst besides simply increasing the amount of vanadium. For example, G. He et al. reported that 1 wt.% V_2O_5/TiO_2 catalyst with large proportions of polymeric vanadyl species can be prepared by using TiO_2 support pretreated with sulfate species (G. He et al., *Sci. Adv.* 4, eaau4637 (2018)). However, our work has mainly focused on solving the physical deactivation by ABS deposition at low temperatures rather than designing catalysts with improved low-temperature activity. In fact, when we prepared several transition metal-based catalysts containing Mn, Ce, Sb and tested them under real conditions containing 5% CO_2 and 10% H_2O , we could not find a catalyst with better low-temperature activity than 5 wt.% V/W- TiO_2 catalyst. For this reason, a high V loading catalyst was used to reach target low-temperature NO_x conversion at low-temperature. However, the novelty of our work comes from mitigating physical deactivation through zeolite mixing rather the use of the high V loading catalyst itself. It should be noted that ABS deactivation also occurs when catalysts with lower V loading are used, but much slower. Therefore, the strategy of trapping ABS using zeolite can still be effectively used for lower V loading catalysts. Future research will be able to expand towards the development of metal oxides-zeolite hybrid catalysts without the use of high V loading catalyst.

Question 3. In Supplementary Fig. 2, we can see that the NO_x conversion of VWTi+Z exceeded 100 % within the range from 250 °C to 300 °C, which is unreasonable. Please recheck the raw data and modify this figure.

Answer: Thank you for your comments. It seems that the NO_x conversion line was plotted to be more than 100% in the processing of connecting the data of each temperature with a spline curve in the Origin program. The actual measured data are all lower than 100% (dot data), which are reasonable. To avoid misinterpretation, the NO_x conversion data are replotted with straight line.

Supplementary Fig. 2. (revised)

Question 5. There were some Arrhenius plots of catalysts obtained in this paper to describe the state of surface species. To avoid the mass transfer limitations, the Arrhenius plots should be measured at low conversions (< 20%) (A. Wang et al., Nat Commun 10, 1137 (2019); J. Song et al., ACS Catal. 7, 12, 8214–8227 (2017)). However, in this work, the NOx conversion of 10 wt.% ABS/VWTi+Z catalyst achieved 20 % at 150 °C while the temperature range of apparent energy tests was from 135 °C to 215 °C which could affect the reliability of some conclusions. What's more, the NOx conversion data of catalysts with 2 wt.% ABS was not provided in this paper, which could be more easily to exceed 20 % of NOx conversion.

Question 8. For the answer to question 6 and 7, the decrease of the activation energy of both samples above 160 °C was explained by the gap between steady state reaction and transient reaction. If this explanation is correct, the discussion around the pre-exponential factor values of catalysts should also be inaccurate. Furthermore, the unperfect linearity Arrhenius plots of VWTi could be caused by the mass transfer limitations at a high conversion which has been mentioned above. Please learn more references and literatures around apparent energy test. These mistakes made in the discussion of key data were extremely serious. Hence, I suggest this paper should be rejected.

Answer: Thank you for your valuable and critical comments. As the reviewer pointed out, the activation energy values of ABS/VWTi+Z and ABS/VWTi+Z PM L (2 wt.% ABS impregnated) were obtained in the range of high NOx conversion above the 20%, which could be affected by the mass transfer limitation. Therefore, to measure the E_a values more precisely, we performed the steady state SCR reactions with ABS/VWTi+Z and

ABS/VWTi+Z PM L. The space velocities of the reaction at low temperature (135-155 °C) and high temperature (195-215 °C) were differently set for controlling the NO_x conversion to be below 20% (120,000 h⁻¹ and 400,000 h⁻¹, respectively). The activation energy of VWTi is 70 and 54 kJ/mol at low and high temperature range, respectively (Figure R1 black).

Figure R1. The Arrhenius plots of VWTi, ABS/VWTi+Z and ABS/VWTi+Z PM L under steady-state SCR reaction condition (500 ppm NO, 600 ppm NH₃, 10% O₂, 5% CO₂, 10% H₂O, N₂ balance). All reaction rates were measured under the reaction condition where the NO_x conversions were below 20%.

Obtained 70 and 54 kJ/mol at low and high temperature range implies that rate determining step of SCR reaction on VWTi is different depending on the temperature. These activation energy values are almost same to the values measured from transient experiment, which were ~70 and ~59 kJ/mol at low and high temperature, respectively (Supplementary Table 2). The reported activation energy was also 57 kJ/mol at the temperature range 160-280 °C in the previous literature (G, Yadolah, et al. Applied Catalysis B: Environmental 2020, 278). In the case of much lower temperature (below 155 °C), the SCR reaction is inhibited by H₂O desorption on the surface, so the rate determining step of reaction shifts from re-oxidation of reduced V sites to the H₂O desorption from V sites at the low temperature (A, Logi, et al. Journal of Catalysis 346:188-197, 2017). Such rate determining step shift can lead to the different E_a in low and high temperature.

In the Arrhenius plots of ABS/VWTi+Z and ABS/VWTi+Z PM L, their reactivity and activation energy in the low temperature region were almost identical to those of VWTi at

low temperature. It means that impregnated ABS has little effect on the SCR reactivity at low temperature because the ABS is solid phase under 160 °C as we suggested in the earlier revision. At high temperature, however, the reactivity decreased in the ABS/VWTi+Z PM L compared to VWTi, which means that the liquid ABS blocked the pores of VWTi and deactivated the catalyst. In this case, activation energy was same in the ABS/VWTi+Z PM L and VWTi, but only pre-exponential factor decreased (parallel blue line beneath the black Arrhenius plot of VWTi in Figure R1) strongly supporting that the deactivation by ABS is physical deactivation. However, the ABS/VWTi+Z demonstrated same Arrhenius plot as VWTi even at high temperature indicating that the deactivation of VWTi vanished via the migration of ABS into Y zeolite. All these results were consistent with our data obtained from transient experiments, indicating that the diffusion limitation or the gap between transient and steady-state reaction can be ignored in our reaction condition.

Supplementary Table 2.

Activation energy (E_a) and pre-exponential factor obtained from the Arrhenius plots of ABS/VWTi+Z and ABS/VWTi+Z PM L during the transient SCR reaction

		< 160 °C	> 180 °C
ABS/VWTi+Z	E_a (kJ/mol)	70.4	61.1
	A	2.1×10^7	1.8×10^6
ABS/VWTi+Z PM L	E_a (kJ/mol)	70.4	58.3
	A	2.0×10^7	6.7×10^5

Question 4. For the answer to question 4, the phenomenon that the apparent activation energy of 10 wt.% ABS/VWTi+Z PM L got changed during ramping was thought to be caused by the excess loading amount of ABS and the on-going migration of ABS. However, there was no relevant reference to support your assumption and the migration of excess ABS was still the physical behavior of deposit on the surface of catalysts, which should not change the apparent activation energy. Please provide some more reference and give a logical explanation.

Answer: We thank the reviewer for the insightful question. In order to illustrate the effect of ABS more precisely, the results of 2wt ABS/VWTi+Z and 2wt ABS/VWTi+Z PM L were shown in Figure 3a (notated as ABS/VWTi+Z and ABS/VWTi+Z PM L in Figure 3a) instead of 10wt ABS/VWTi+Z and 10wt ABS/VWTi+Z PM L. As liquid ABS results in the physical deactivation, the Arrhenius plot of ABS/VWTi+Z PM L should have same curve to that of ABS/VWTi+Z at low temperature (below 160 °C) and parallel curve beneath that of ABS/VWTi+Z at high temperature (above 180 °C), which was discussed in earlier revision. Such behavior was clearly demonstrated in Arrhenius plots of ABS/VWTi+Z and ABS/VWTi+Z PM L (Figure 3a, Figure R2). However, in 10wt ABS/VWTi+Z and 10wt

ABS/VWTi+Z PM L, there are some mismatched points to the model that we assume, such as different E_a at high temperature (Supplementary Fig.11). We can easily realize that there should be some transient regime between low and high temperature in the Arrhenius plot (Figure R2). It is a regime where the impregnated ABS starts to melt, and deactivate the VWTi or migrate into Y zeolite. Therefore, we should avoid such regime to conduct the kinetic analysis. If the amount of ABS was too much in the ABS/VWTi+Z and ABS/VWTi+Z PM L (ABS =10wt%), it is expected that such transient regime was much wider, which hinders the exact kinetic analysis. Unfortunately, we could not find relevant reference which supports our description, but we think that it is logical enough to be generally accepted.

Revised manuscript

Figure R2. The Arrhenius plots of ABS/VWTi+Z and ABS/VWTi+Z PM L (2wt% ABS) under transient reaction (500 ppm NO, 600 ppm NH₃, 10% O₂, 5% CO₂, 10% H₂O, N₂ balance) with ramp rate of 1 °C /min.

Figure 3a (revised). Arrhenius plots of ABS/VWTi+Z (black dot) and ABS/VWTi+Z PM L (red dot) during transient NH₃-SCR reaction (temperature from 100 °C to 220 °C with ramp rate 1 °C/min)

Supplementary Fig. 11 (revised). Arrhenius plots of 10 wt.% ABS/VWTi+Z (black dot) and 10 wt.% ABS/VWTi+Z PM L (red dot) during transient NH₃-SCR reaction (temperature from 100 °C to 220 °C with ramp rate 1 °C/min, other reaction conditions were the same as steady-

state NH_3 -SCR experiment. The 10 wt.% ABS/VWTi sample was prepared by wet impregnation method of 10 wt.% of ABS on the VWTi catalyst.

Supplementary Fig. 10 (revised)

Comparison of physically mixed VWTi and Y zeolite (VWTi+Z) to loose contact sample (VWTi+Z PM L).

a, NOx conversion data of ABS/VWTi+Z and PM L were obtained from transient NH_3 -SCR reaction at temperature from 100 to 220 °C with ramp rate 1 °C/min. ABS/VWTi+Z PM L had reactivity of 45 % NOx conversion at 220 °C indicating deactivation by impregnated ABS. In contrast, ABS/VWTi+Z showed almost same activity to fresh VWTi (NOx conversion 59% at 220 °C). It means that physical contact mitigated the deactivation of VWTi which was blocked by impregnated ABS. **b**, 22 h SO_2 aging profiles of VWTi+Z PM L under NH_3 -SCR reaction condition with 30 ppm SO_2 at 220 °C. NH_3 -SCR condition was under 500 ppm NO, 600 ppm NH_3 , 10% O_2 , 5% CO_2 , 10% H_2O and balance N_2 . GHSV: 150,000 mL/h·g_{cat} based on VWTi weight.

Question 6. There had the same issue on all figures of Arrhenius plots such as Fig. R1, Fig.

R2 and Supplementary Fig. 11 that the scale of the bottom cannot be in line with that of the top. For example, for $1000/K=2.1$, the corresponding centigrade temperature should be 203 °C which was on the left of 200 °C instead of on the right of 200 °C on the top of figure. Please recheck all the pictures carefully and modify the description of these figure in this paper.

Answer: Thank you for your comments. As a reviewer pointed out, there was some error in top X axis in the process of changing the unit K to °C. As the axis is unnecessary, we modify the figure deleting the top axis.

Revised manuscript:

Figure 3a (revised).

Supplementary Fig. 11 (revised).

Question 7. For the answer to question 6 and 7 and the revised manuscript, the name of catalysts were 2wt ABS/VWTi+Y and 2wt ABS/VWTi+Y PM L. However, the catalysts in original text were named as 2wt ABS/VWTi+Z and 2wt ABS/VWTi+Z PM L, which should be rechecked and modified.

Answer: Thank you for your comments. We recheck the all figure and correct the name of the catalyst to VWTi+Z as described in the original text.

Revised manuscript:

Supplementary Fig. 4. Reaction test under SO₂ in higher concentration. Reaction profiles of 22 h SO₂ aging over VWTi+Z under NH₃-SCR condition with different SO₂ concentration at 220 °C (500 ppm NO, 600 ppm NH₃ 10% O₂, 5% CO₂, 10% H₂O, 30 ppm or 100 ppm SO₂ and balanced with N₂, GHSV : 150,000 mL/h·g_{cat} based on VWTi weight).

Supplementary Fig. 14. Characteristics of Y zeolite and Sulfated Y zeolite. Although the sulfate was bound to framework, there was little change in the Brønsted acid sites of Y zeolite. It might be because the sulfated sites can also act as the Brønsted acid sites. This results in the almost same sulfur resistance performance of VWTi+Z catalyst after regeneration.

REVIEWER COMMENTS

Reviewer #1 (Remarks to the Author):

In the latest response, although authors have provided explanation and supplemented some more experimental data to address my questions, the significance and guidance to this field of this work cannot be satisfactory. Furthermore, some unsolved issues still exist in this article. All in all, this paper cannot be accepted.

1. As known to all, the defects of commercial V based catalysts mainly concentrate on the poor low-temperature activity, inevitable biotoxicity, and high oxidation of SO₂. In this work, to get higher activity at low temperature, vanadia/titania catalysts with a higher loading amount (5 wt. %) of V₂O₅ was adopted. Although the catalytic performance got acceptable, thermal stability, biotoxicity, and SO₂ oxidation got much more serious as well (He et al., *Sci. Adv.* 4, eaau4637 (2018)). Despite some high-level works mention that the increasing active sites of V based catalysts can promote the low-temperature activity (He et al., *Sci. Adv.* 4, eaau4637 (2018); Jaegers, N.R. et al., *Angew. Chem. Int. Ed.* 131, 12739-12746 (2019); Nielsen, U.G. et al., *J. Am. Chem. Soc.* 126, 4926-4933 (2004)), all these works are aimed at exploring the specific active site, polymeric vanadyl species, to improve catalytic performance of catalysts with a fewer loading of V₂O₅ but not guide us to increase the loading of V₂O₅ simply. Hence, the research of V-based catalysts should focus on developing catalysts with highly uniform oligomeric vanadia sites in the premise of lower amount of V₂O₅. And I firmly insist that the significance and guidance of this work cannot meet the level of this journal.

2. For solving SO₂-poisoning SCR catalysts, as mentioned in your response to Question 1, this work does not lay the roots in the SO₂ adsorption and oxidation which are the root problems. I totally understand the irreplaceability of VWTi catalyst in burn fuels containing sulfur for the moment. However, for work qualified to be published in this journal, it should be the bellwether of this field and thus physically mixing zeolites to lower the regeneration temperature of SO₂ aged VWTi, which has certain industrial significance but is short of revolutionary scientific significance and even could mislead further research in developing SO₂-resistant SCR catalysts, should not be proposed in this journal.

3. For solving ABS poisoning of V based catalysts, physically mixing zeolites lowered the regeneration temperature in this work. However, due to the enhanced SO₂ oxidation by higher loading of V₂O₅, more and more ABS could deposit in the pipeline which needs to devote more cost and research. In the response to Question 1, you said that the top priority of our research is to prevent the rapid deactivation of SCR catalysts by SO₃. But the most meaningful highlight of this work lays emphasis on industrial application that solving the urgent environmental issues faced. Thus, the industrial issues like enhanced SO₂ oxidation and anabatic pipeline corrosion brought by the higher loading of V₂O₅ must be taken into consideration.

4. In this work, physically mixed H-Y zeolite was thought to hardly capture SO₂ under SO₂ aging condition (line 98 - 99), which was deduced by Supplementary Fig. 3a. However, there was an obvious peak around 900 °C for Y zeolite-SO₂ aged. This result showed that some stable sulfates deposited on SO₂ aged samples, which could be aluminum sulfate demonstrated in another section below of this article. Thus the result above was incorrect. Moreover, VWTi+Silica-SO₂ aged was also tested in Supplementary Fig. 3a to prove that H-Y zeolite only prevented ABS deactivation and did not capture

SO₂. Whereas VWTi + Silica is not equal to VWTi + Y zeolite. Thus, the mass spectrometry results of SO₂ aged VWTi + Silica should not be used to evaluate the role of VWTi+ Y zeolite.

5. The amount of deposited sulfur species over SO₂ aged catalysts was analyzed (Supplementary Table 1). However, the deposited amount of S of VWTi SO₂ aged was smaller than that of VWTi+Z SO₂ aged, which meant that the mixture of zeolite promoted the deposition of sulfur species after SO₂ aged.

Please give the relevant explanation for this phenomenon.

6. As mentioned in this work, the phenomenon that initial locations where sulfur species formed were different from the regions where they were deposited in the VWTi+Z was thought to be not caused by a simple diffusion process. However, the two relevant reasons that i) nearly no sulfur remained on VWTi compared to zeolite regions in the VWTi+Z mixed catalyst, and ii) almost no change in ABS location was observed for aged VWTi+Silica mixtures cannot support your deduction above directly. Please give more reasonable and detailed explanation.

7. In the response to Question 8, according to the previous suggestion, E_a values were measured more precisely after the space velocities at high temperature got controlled. And, as said in this response, the latest results were consistent with data obtained from transient experiments (Supplementary Table 2) so that the diffusion limitation or the gap between transient and steady-state reaction can be ignored in your reaction condition. However, the data listed in Supplementary Table 2 was got from the experiments on 10 wt % ABS/samples (line 180-185) and E_a values in the response was measured on 2 wt % ABS/samples. These two data sources were totally different! What's more, compared with the Arrhenius plots of ABS/VWTi+Z and ABS/VWTi+Z PM L (2wt% ABS) (Figure R2) that the slope of ABS/VWTi+Z was unchanged within the testing temperature window. It was totally inconsistent with the results in this response. Thus, the content must be modified carefully.

8. In the response to Question 8, you thought that the diffusion limitation or the gap between transient and steady-state reaction can be ignored in your reaction condition. Even if it really occurs in this catalytic system, the diffusion limitation and the gap between transient and steady-state reaction should still be taken into consideration and update relevant data in this paper because this phenomenon could just be an incidental in this work but could not happen in other catalytic systems. Thus if this paper maintains like this version, the measurement of E_a in further research will be misled.

Response to the reviewer's comments

Sincerest thanks for your response and referee's valuable advice and comments on our manuscript. A point by point response to the reviewer's comments were attached below, marked in blue, to fully address the reviewer's comments. We hope that the revisions will comply with the referee's remarks.

[Reviewer #1]

In the latest response, although authors have provided explanation and supplemented some more experimental data to address my questions, the significance and guidance to this field of this work cannot be satisfactory. Furthermore, some unsolved issues still exist in this article. All in all, this paper cannot be accepted.

Question 1. As known to all, the defects of commercial V based catalysts mainly concentrate on the poor low-temperature activity, inevitable biotoxicity, and high oxidation of SO₂. In this work, to get higher activity at low temperature, vanadia/titania catalysts with a higher loading amount (5 wt.%) of V₂O₅ was adopted. Although the catalytic performance got acceptable, thermal stability, biotoxicity, and SO₂ oxidation got much more serious as well (He et al., Sci. Adv. 4, eaau4637 (2018)). Despite some high-level works mention that the increasing active sites of V based catalysts can promote the low-temperature activity (He et al., Sci. Adv. 4, eaau4637 (2018); Jaegers, N.R. et al., Angew. Chem. Int. Ed. 131, 12739-12746 (2019); Nielsen, U.G. et al., J. Am. Chem. Soc. 126, 4926-4933 (2004)), all these works are aimed at exploring the specific active site, polymeric vanadyl species, to improve catalytic performance of catalysts with a fewer loading of V₂O₅ but not guide us to increase the loading of V₂O₅ simply. Hence, the research of V-based catalysts should focus on developing catalysts with highly uniform oligomeric vanadia sites in the premise of lower amount of V₂O₅. And I firmly insist that the significance and guidance of this work cannot meet the level of this journal.

Answer: We thank the reviewer for valuable comments. Although the reviewer consistently pointed out the high loading V₂O₅, our work is focused on solving ABS deactivation problem as well as achieving high activity at low-temperatures. The message and guidance of our work is 'Strategy of trapping ABS by utilizing physically mixed zeolite can enable SCR operation at low-temperatures with catalysts having high activity', rather than 'Simply increasing V loading can enhance SCR activity at low-temperatures'. We fully agree that future research is needed to develop V-based SCR catalysts with lower V loading, however, we also believe that our ABS trapping strategy can be acceptable and compatible with future research directions.

In addition, it should be noted that the strategy of trapping ABS using zeolite can still be effectively used for lower V loading catalysts. In this revision, we compared the SCR performance of 3 wt.% V₂O₅/WO₃-TiO₂ catalyst (3VWTi) and its mixture with Y zeolite

(3VWTi+Z) under 100 ppm SO₂ for accelerated aging (Supplementary Fig. 7). Although initial activity decreases to ~30% on 3VWTi, the ABS deactivation also occurs when catalyst with lower V loading are used, but much slower when Y zeolite is physically mixed. It was obviously confirmed that ABS trapping function of zeolite works consistently when mixed with lower V loaded catalyst as well as high V loaded catalyst. Such observations sufficiently demonstrate the potential of this approach for further application to other general SCR catalysts with various vanadia loadings in the industry since slower deactivation implies fewer regeneration, which leads to provide financial (less energy consumption) as well as environmental (less CO₂ emission) benefits significantly.

Supplementary Fig. 7.

Reaction tests under 100 ppm SO₂ for VWTi catalyst with 3 wt.% V₂O₅.

Reaction profiles of 22 h SO₂ aging over 3 wt.% V₂O₅/WO₃-TiO₂ catalyst (3VWTi) and its mixture with Y zeolite (3VWTi+Z) under NH₃-SCR condition at 220 °C (500 ppm NO, 600 ppm NH₃ 10% O₂, 5% CO₂, 10% H₂O, 100 ppm SO₂ and balanced with N₂, GHSV : 150,000 mL/h·g_{cat} based on VWTi weight).

Revised manuscript (line 106): Furthermore, VWTi catalyst with a much lower amount of V (3 wt.% V₂O₅) was also compared with its mixture with zeolite under SO₂ aging condition (Supplementary Fig. 7), verifying that the mixed zeolite can prevent ABS deactivation regardless of V loading.

Question 2. For solving SO₂-poisoning SCR catalysts, as mentioned in your response to Question 1, this work does not lay the roots in the SO₂ adsorption and oxidation which are the

root problems. I totally understand the irreplaceability of VWTi catalyst in burn fuels containing sulfur for the moment. However, for work qualified to be published in this journal, it should be the bellwether of this field and thus physically mixing zeolites to lower the regeneration temperature of SO₂ aged VWTi, which has certain industrial significance but is short of revolutionary scientific significance and even could mislead further research in developing SO₂-resistant SCR catalysts, should not be proposed in this journal.

Answer: Thank you for the comments. The modern chemical society is beginning to take interest in the mechanochemistry that occurs when two or more materials are mechanically mixed beyond single phase material (A. P. Amrute et al., *Science* **25**, 485-489 (2019); K. Ralphs et al., *Chem. Soc. Rev.* **42**, 7701-7718 (2013)). Mechanical mixing gives a rise to synergistic effect between the materials. Therefore, resulting physical mixtures have great potential in catalytic reactions depending on the combination of various catalysts, however, few studies have been reported to date. In this context, our report is a pioneering work to overcome deactivation by utilizing physical mixture for ABS trapping, which is attributed to the different interaction between deactivating substance and mechanical mixture consisting of V catalyst and zeolite Y. Therefore, our work has deep significance in encompassing two keywords in future catalyst research: mechanochemistry and catalyst deactivation.

Question 3. For solving ABS poisoning of V based catalysts, physically mixing zeolites lowered the regeneration temperature in this work. However, due to the enhanced SO₂ oxidation by higher loading of V₂O₅, more and more ABS could deposit in the pipeline which needs to devote more cost and research. In the response to Question 1, you said that the top priority of our research is to prevent the rapid deactivation of SCR catalysts by SO₃. But the most meaningful highlight of this work lays emphasis on industrial application that solving the urgent environmental issues faced. Thus, the industrial issues like enhanced SO₂ oxidation and anabatic pipeline corrosion brought by the higher loading of V₂O₅ must be taken into consideration.

Answer: Thank you for the valuable question. As we mentioned in the previous revision, the reason for potential pipeline corrosion is that vanadia catalyst oxidizes SO₂ to SO₃ and then SO₃ is released downstream. In our zeolite-mixed catalyst system, most of generated SO₃ can be trapped in the form of ABS in zeolite based on the quantification results of residual sulfur on aged catalyst. The only problem in the system is the instantaneous release of stored SO₃ during the regeneration, however, other technical solutions such as installing wet electrostatic precipitator or injecting an alkaline solvent in a targeted manner can be alternatively considered in the previous study (R. K. Srivastava et al., *J. Air & Waste Manage. Assoc.* 54:750-762 (2004); C. Zheng et al., *Fuel* 259, 116306 (2020); H. Zhao et al., *Aerosol Air Qual. Res.* 18:2906-2911 (2018); Z. Yang et al., *Fuel* 217, 597-604 (2018)).

Question 4. In this work, physically mixed H-Y zeolite was thought to hardly capture SO₂ under SO₂ aging condition (line 98-99), which was deduced by Supplementary Fig. 3a.

However, there was an obvious peak around 900 °C for Y zeolite-SO₂ aged. This result showed that some stable sulfates deposited on SO₂ aged samples, which could be aluminum sulfate demonstrated in another section below of this article. Thus the result above was uncorrect. Moreover, VWTi+Silica-SO₂ aged was also tested in Supplementary Fig. 3a to prove that H-Y zeolite only prevented ABS deactivation and did not capture SO₂. Whereas VWTi + Silica is not equal to VWTi + Y zeolite. Thus, the mass spectrometry results of SO₂ aged VWTi + Silica should not be used to evaluate the role of VWTi+ Y zeolite.

Figure R1. Thermal decomposition profiles of sulfur species on the SO₂-aged VWTi+Z and Y zeolite. (Data were replotted from Fig. 2f and Supplementary Fig. 3a in the manuscript.)

Answer: Thanks for the reviewer’s comments. As reviewer pointed out, a small amount of aluminum sulfate was observed to be formed on Y zeolite during exposure to SO₂. However, this is relatively small amount (1.1×10^{-10}) compared to the amount of aluminum sulfate in VWTi+Z-SO₂ aged catalyst (1.3×10^{-9} , integrated value from Figure R1). Such observations demonstrate that the formation of SO₃ on VWTi catalyst is prerequisite for the formation of aluminum sulfate in zeolite. VWTi+Silica was tested for comparison with VWTi+Y zeolite to investigate the specific role of zeolite in preventing ABS deactivation. Unlike Y zeolite, physically mixed silica cannot trap ABS during low-temperature SCR operation, demonstrating that the interaction between ABS and physically-mixed material determines ABS trapping ability. Furthermore, most of sulfur species exist in the form of ABS based on the mass spectrometry results of SO₂-aged VWTi+Silica, and no peaks of metal sulfate are observed, which is also well-matched with the reaction results mentioned above.

Question 5. The amount of deposited sulfur species over SO₂ aged catalysts was analyzed (Supplementary Table 1). However, the deposited amount of S of VWTi SO₂ aged was smaller than that of VWTi+Z SO₂ aged, which meant that the mixture of zeolite promoted the deposition of sulfur species after SO₂ aged. Please give the relevant explanation for this phenomenon.

Answer: Thanks for the important comment. As the reviewer pointed out, the amount of sulfur on the VWTi+Z is larger than that on VWTi. This is due to different deactivation rates of the two catalysts. When ABS deactivation occurs on VWTi, the SO₂ oxidation activity as well as the SCR activity gradually decreases as the vanadium active site is covered with ABS. In the case of VWTi+Z, however, ABS deactivation hardly occurred, so it is natural that more SO₂ was oxidized to SO₃ and more sulfur species were deposited on the catalyst during 22 h aging experiment.

Question 6. As mentioned in this work, the phenomenon that initial locations where sulfur species formed were different from the regions where they were deposited in the VWTi+Z was thought to be not caused by a simple diffusion process. However, the two relevant reasons that i) nearly no sulfur remained on VWTi compared to zeolite regions in the VWTi+Z mixed catalyst, and ii) almost no change in ABS location was observed for aged VWTi+Silica mixtures cannot support your deduction above directly. Please give more reasonable and detailed explanation.

Answer: We appreciate the reviewer's comment. We think that above two observations can clearly demonstrate our deduction. First, if we assume that the ABS migration is simple diffusion process mediated by ABS concentration gradients, the amount of ABS on VWTi particle and zeolite particle should be similar. However, nearly no sulfur remained on VWTi compared to zeolite regions in the VWTi+Z catalyst, implying that this phenomenon occurred by chemical interactions. Second, if we assume that the ABS migration is simple diffusion process mediated by ABS concentration gradients, formed ABS may also diffuse into silica particles (~285 m²/g). However, almost no change in ABS location was observed for aged VWTi+Silica mixtures, clearly demonstrating that ABS migration is derived by chemical interaction rather than simple physical diffusion. Such hypothesis is also confirmed by the results of theoretical calculation, as shown in the manuscript (Fig. 3b).

Question 7. In the response to Question 8, according to the previous suggestion, E_a values were measured more precisely after the space velocities at high temperature got controlled. And, as said in this response, the latest results were consistent with data obtained from transient experiments (Supplementary Table 2) so that the diffusion limitation or the gap between transient and steady-state reaction can be ignored in your reaction condition. However, the data listed in Supplementary Table 2 was got from the experiments on 10 wt % ABS/samples (line 180-185) and E_a values in the response was measured on 2 wt % ABS/samples. These two data sources were totally different! What's more, compared with the Arrhenius plots of ABS/VWTi+Z and ABS/VWTi+Z PM L (2wt% ABS) (Figure R2) that the slope of ABS/VWTi+Z was unchanged within the testing temperature window. It was totally inconsistent with the results in this response. Thus, the content must be modified carefully.

Supplementary Table 2.

Activation energy (E_a) and pre-exponential factor obtained from the Arrhenius plots of ABS/VWTi+Z and ABS/VWTi+Z PM L (2wt.% ABS) during the transient SCR reaction

		< 160 °C	> 180 °C
ABS/VWTi+Z	E_a (kJ/mol)	70.4	61.1
	A	2.1×10^7	1.8×10^6
ABS/VWTi+Z PM L	E_a (kJ/mol)	70.4	58.3
	A	2.0×10^7	6.7×10^5

Figure R2. The Arrhenius plots of VWTi, ABS/VWTi+Z and ABS/VWTi+Z PM L (2wt.% ABS) under steady-state SCR reaction condition (500 ppm NO, 600 ppm NH₃, 10% O₂, 5% CO₂, 10% H₂O, N₂ balance). All reaction rates were measured under the reaction condition where the NO_x conversions were below 20%.

Answer: Thanks for your comment. First of all, there might be some misunderstanding due to a lack of explanation because the data in Supplementary Table 2 are exactly the activation energy value of ABS/VWTi+Z and ABS/VWTi+Z PM L (2wt.% ABS, not 10wt.% ABS) calculated from the data in Fig. 3a. Therefore, data from Figure R2 and Supplementary Table 2 can be directly compared. In addition, the slope of ABS/VWTi+Z in Fig. 3a is also different in low and high temperature regime like ABS/VWTi+Z PM L. This can be confirmed by the E_a data in Supplementary Table 2, which is calculated from the data in Fig. 3a. Meanwhile,

we conducted same experiments with much higher amount of pre-impregnated ABS (10 wt.%) on VWTi (Supplementary Fig. 12), and the difference between the two samples was more clearly observed. As mentioned in the previous revision, however, a transient regime still exists up to high temperature for the case of '10wt.% ABS/VWTi+Z PM L' sample in the Arrhenius plot (Supplementary Fig. 12). It is a regime where the impregnated ABS starts to melt, and deactivate the VWTi or migrate into Y zeolite. We think that if the amount of ABS was too much in the ABS/VWTi+Z and ABS/VWTi+Z PM L (ABS =10wt.%), such transient regime was much wider, which hinders the exact kinetic analysis. That is why we only calculate the activation energy for 2wt.% ABS impregnated catalyst in Supplementary Table 2 to avoid such a transient regime.

Supplementary Fig. 12.

Arrhenius plots of 10wt ABS/VWTi+Z (black dot) and 10wt ABS/VWTi+Z PM L (red dot) during transient NH₃-SCR reaction (temperature from 100 °C to 220 °C with ramp rate 1 °C/min, other reaction conditions were the same as steady-state NH₃-SCR experiment. The 10wt ABS/VWTi sample was prepared by wet impregnation method of 10 wt.% of ABS on the VWTi catalyst.

Revised manuscript (line 188): Note that we conducted same experiments with much higher amount of pre-impregnated ABS (10wt.%) on VWTi (Supplementary Fig. 12), and the difference between the two samples was remarkably observed.

Question 8. In the response to Question 8, you thought that the diffusion limitation or the gap

between transient and steady-state reaction can be ignored in your reaction condition. Even if it really occurs in this catalytic system, the diffusion limitation and the gap between transient and steady-state reaction should still be taken into consideration and update relevant data in this paper because this phenomenon could just be an incidental in this work but could not happen in other catalytic systems. Thus if this paper maintains like this version, the measurement of E_a in further research will be misled.

Figure R2. The Arrhenius plots of VWTi, ABS/VWTi+Z and ABS/VWTi+Z PM L (2wt.% ABS) under steady-state SCR reaction condition (500 ppm NO, 600 ppm NH₃, 10% O₂, 5% CO₂, 10% H₂O, N₂ balance). All reaction rates were measured under the reaction condition where the NO_x conversions were below 20%.

Figure R3. The Arrhenius plots of ABS/VWTi+Z and ABS/VWTi+Z PM L (2wt.% ABS) under transient reaction (500 ppm NO, 600 ppm NH₃, 10% O₂, 5% CO₂, 10% H₂O, N₂ balance) with ramp rate of 1 °C /min.

Answer: Thank you for the comment. We also agree that the reaction kinetic should be carefully analyzed. Therefore, in the experiment of Figure R2, the reactions were conducted under steady-state condition with low NO_x conversion (< 20%) to precisely measure the activation energy reflecting the previous comment of the reviewer. As a result, the obtained value from Figure R2 was almost consistent with that from Figure R3 (Fig. 3a in the manuscript) which was a transient reaction. Based on such results, it is reasonable to conclude that the gap between the transient reaction in Figure R3 and steady-state reaction in Figure R2 was almost diminished in our reaction condition. We did not intend to generalize this interpretation in all catalytic system, just describing this conclusion in ‘our system’ based on the obtained data thoroughly. In addition, we also obtained the value of activation energy of ~54 kJ/mol above 160 °C for VWTi catalyst under steady-state SCR condition, which is consistent with recent publication that obtained 55~59 kJ/mol at temperatures between 160-280 °C (G, Yadolah, et al. Applied Catalysis B: Environmental 2020, 278). (The higher activation energy of ~70 kJ/mol was only obtained at very low temperatures between 135-160 °C.) In the case of much lower temperature, the SCR reaction is inhibited by H₂O desorption on the surface, so the rate determining step of reaction can shift from re-oxidation of reduced V sites to the H₂O desorption from V sites at the low temperature (A, Logi, et al. Journal of Catalysis 346:188-197, 2017). We suppose that such shift of rate determining step can lead to the different E_a in low and high temperature. Therefore, our results are consistent with the previous report, and there is no reason to judge that the kinetic analysis in our further researches will be misled.